# Exploiting pyocyanin to treat mitochondrial disease due to respiratory complex III dysfunction

Roberta Peruzzo [1,8], Samantha Corrà[1], Roberto Costa[1], Michele Brischigliaro [1], Tatiana Varanita[1], Lucia Biasutto[2,3], Chiara Rampazzo [1], Daniele Ghezzi [4,5], Luigi Leanza [1], Mario Zoratti[2,3], Massimo Zeviani [6,7], Cristiano De Pittà [1], Carlo Viscomi [3], Rodolfo Costa [1] & Ildikò Szabò [1,2 ✉]

Mitochondrial diseases impair oxidative phosphorylation and ATP production, while effective treatment is still lacking. Defective complex III is associated with a highly variable clinical spectrum. We show that pyocyanin, a bacterial redox cycler, can replace the redox functions of complex III, acting as an electron shunt. Sub-µM pyocyanin was harmless, restored respiration and increased ATP production in fibroblasts from five patients harboring pathogenic mutations in *TTC19*, *BCS1L* or *LYRM7*, involved in assembly/stabilization of complex III. Pyocyanin normalized the mitochondrial membrane potential, and mildly increased ROS production and biogenesis. These in vitro effects were confirmed in both *Drosophila*[TTC19KO] and in *Danio rerio*[TTC19KD], as administration of low concentrations of pyocyanin significantly ameliorated movement proficiency. Importantly, daily administration of pyocyanin for two months was not toxic in control mice. Our results point to utilization of redox cyclers for therapy of complex III disorders.

[1] Department of Biology, University of Padova, Padova, Italy. [2] CNR Institute of Neuroscience, Padova, Italy. [3] Department of Biomedical Sciences, University of Padova, Padova, Italy. [4] Unit of Medical Genetics and Neurogenetics, Fondazione IRCCS Istituto Neurologico Carlo Besta, Milano, Italy. [5] Department of Pathophysiology and Transplantation, University of Milan, Milan, Italy. [6] Department of Neurosciences, University of Padova, Padova, Italy. [7] Venetian Institute of Molecular Medicine, Padova, Italy. [8] Present address: Department of Molecular and Cell Biology, University of California, Berkeley, CA, USA. ✉email: ildiko.szabo@unipd.it

Mitochondria carry out the synthesis of most of the ATP needed for cellular functions. A defect in ATP production is usually more deleterious for high-energy-demanding tissues, such as muscle and brain. Reduction in ATP synthesis can be due to impaired function of the components of the respiratory chain (RC), namely complexes I (CI), II (CII), III (CIII), IV (CIV), and ATP synthase (complex V), which are involved in oxidative phosphorylation (OXPHOS). The energy generated by the electron transport through RC is exploited by CI, CIII, and CIV to pump protons across the inner mitochondrial membrane (IMM) against the electrochemical gradient. The proton-motive force thus generated then drives ATP synthesis through re-entry of protons along the electrochemical gradient, which promotes the rotational activation of complex V. The proton gradient may also carry out other relevant processes such as the translocation of small cations, while the transmembrane potential is required for the transport of proteins across the IMM into the matrix.

Mitochondrial disorders are genetic defects affecting OXPHOS by disrupting the complexes of the RC or ATP synthase, or by affecting mitochondrial DNA (mtDNA) replication/transcription/translation, the biosynthesis of RC cofactors, and mitochondrial biogenesis[1]. Inhibition of electron flow leads not only to OXPHOS impairment with reduced ATP synthesis but also to excess production of reactive oxygen species (ROS) at CI, CII, and/or CIII, as the quinone pool becomes over-reduced[2]. Excess ROS production and limitations of NADH re-oxidation and substrate flux may result in further metabolic impairment, which characterizes the overall pathophysiology of OXPHOS diseases[3]. However, modest, physiological ROS production is a relevant homeostatic mechanism, involved in mitohormesis[4].

Patients with CIII dysfunction develop progressive neurological impairment and meet an early death[5]. Mutations in several genes have been associated with CIII deficiency[1,2,6]. These include mutations in cytochrome $b$ (cyt $b$), the only subunit encoded by mtDNA[7–10], causing myopathy and exercise intolerance and mutations in nine nuclear genes. The most common are nonsense and missense mutations in three factors, namely *TTC19* (tetratricopeptide repeat domain-containing protein 19)[11], *LYRM7*[12], and *BCS1L*[13], which cause defective CIII assembly/stability and function. All these mutations lead to decreased ubiquinol:cyt $c$ oxidoreductase activity. Nonsense mutations in the gene encoding for the TTC19 are associated with impaired stability of the UQCRFS1 (Rieske) electron-transfer protein[14], an iron–sulfur (2S–2Fe) containing catalytic redox protein, resulting in the reduction of mitochondrial respiration by ~40%. This is in turn associated with a spectrum of neurological impairment[11], including progressive encephalomyopathy[15], severe psychiatric disorders[16], Leigh syndrome[17], and cerebellar ataxia[18,19]. LYRM7 chaperones bind the Rieske protein before its incorporation as the last step of the biogenesis of the nascent CIII dimer (CIII2), operated by BCS1L. BCS1L is a member of the AAA+ (ATPases associated with diverse cellular activities) family. Technical diagnostic hurdles are a likely cause for underdiagnosis of CIII-defective patients, relative to those affected by defects in other OXPHOS complexes.

A small number of animal models recapitulating CIII diseases have been utilized to study their pathophysiology. A *dTTC19*-null *Drosophila melanogaster* model shows CIII deficiency associated with strong neurological impairment in adult individuals[11]. A *TTC19* knockdown (KD) zebrafish model displays a significant alteration of embryo morphology[20]. A recently obtained *Ttc19* knockout (KO) mouse shows an ~50% reduction in CIII activity and a progressive neurodegenerative phenotype with motor deficit[21]. A knock-in mouse model has also been generated, harboring the *Bcs1l* c.232A > G mutation, equivalent to the

GRACILE-associated founder mutation identified in Finns and Scandinavians. This mouse displays a severe multisystem phenotype and usually dies within 6 weeks after birth[22].

Recent reports describe results obtained by expressing the alternative oxidase (AOX) from *Ciona intestinalis* in cells with CIII and CIV (cyt $c$ oxidase) deficiency[23,24]. AOX, which is absent in mammals, is a single protein enzyme that bypasses the cyt pathway by shuttling electrons directly from the quinone pool to molecular oxygen, thus potentially overcoming cellular defects associated with CIII/CIV impairment. Because AOX does not pump protons across the IMM, it does not rescue ATP production, but it may unblock electron flow at the level of CIII/CIV. AOX expression has thus been proposed to increase ATP production in CIII- or CIV-defective cells by restoring proton pumping at CI[23]. Accordingly, AOX expression in the mouse model for CIII-linked GRACILE syndrome was shown to prevent lethal cardiac myopathy by increasing cellular respiration[25]. However, the field is controversial, since experiments carried out on a muscle-specific *Cox15$^{−/−}$* mouse model, characterized by a severe cyt $c$ oxidase deficiency in the skeletal muscle, showed marked clinical worsening upon AOX expression, due to impairment of ROS-dependent pathways controlling mitochondrial biogenesis and satellite-cell differentiation[26].

Other treatments aimed at increasing mitochondrial biogenesis by stimulating the peroxisome proliferator-activated receptor gamma coactivator 1α (PGC-1α)-dependent pathway were beneficial in different mouse models of mitochondrial disease due to CIV dysfunction or in Deletor mouse[27–29]. These include the AMPK agonist AICAR, which acts by activating PGC-1α via phosphorylation, and the $NAD^+$ precursor nicotinamide riboside (NR), which stimulates sirtuin1, the deacetylase that activates PGC-1α. However, NR was unable to rescue mitochondrial respiration and prolong the life span in CIII-deficient animals[30], while a ketogenic diet had some modest benefit[31].

Likewise, small molecules able to "replace" defective CIII redox function might ultimately lead to increased ATP production under CIII-defective conditions. We focused on the so-called redox cyclers, compounds undergoing reduction to form radical species that can then react with oxygen, thus regenerating the parent molecule. Thus, we focused on pyocyanin (PYO), a molecule secreted from *Pseudomonas aeruginosa*, because its standard electrochemical potential is close to that of ubiquinol/ubiquinone couple[32], suggesting that PYO may accept electrons from ubiquinol. Indeed, PYO has been suggested to act as an electron shuttle in bacterial respiration. Furthermore, the low molecular weight and zwitterionic properties of PYO may allow the molecule to easily permeate cell membranes and cross the blood–brain barrier[33]. High PYO concentrations (50–100 μM) may cause oxidative stress and be toxic[34].

Here, we provide evidence that very low, nontoxic PYO concentrations can instead increase respiration and ameliorate the phenotypes observed in *TTC19*-defective fruit flies and zebrafish.

## Results

**Effects of PYO in MEFs lacking the CIII-stabilizing factor TTC19.** We first used mouse embryonic fibroblasts (MEFs) from *Ttc19$^{−/−}$* mice[21]. Since in our previous study no differences were observed between *Ttc19$^{+/+}$* and *Ttc19$^{+/−}$* animals for the investigated parameters[21], we used *Ttc19$^{+/−}$* MEFs as controls. At high concentrations (50–100 μM range), similar to those found in *Pseudomonas* infections, PYO is toxic, owing to its redox-active nature, since it can donate electrons not only to cyt $c$ but also to molecular oxygen. Thus, high concentrations of PYO lead to ROS formation and eventually ATP depletion and cell death[32,35,36]. Therefore, we first established the dose response in *Ttc19$^{+/−}$* and

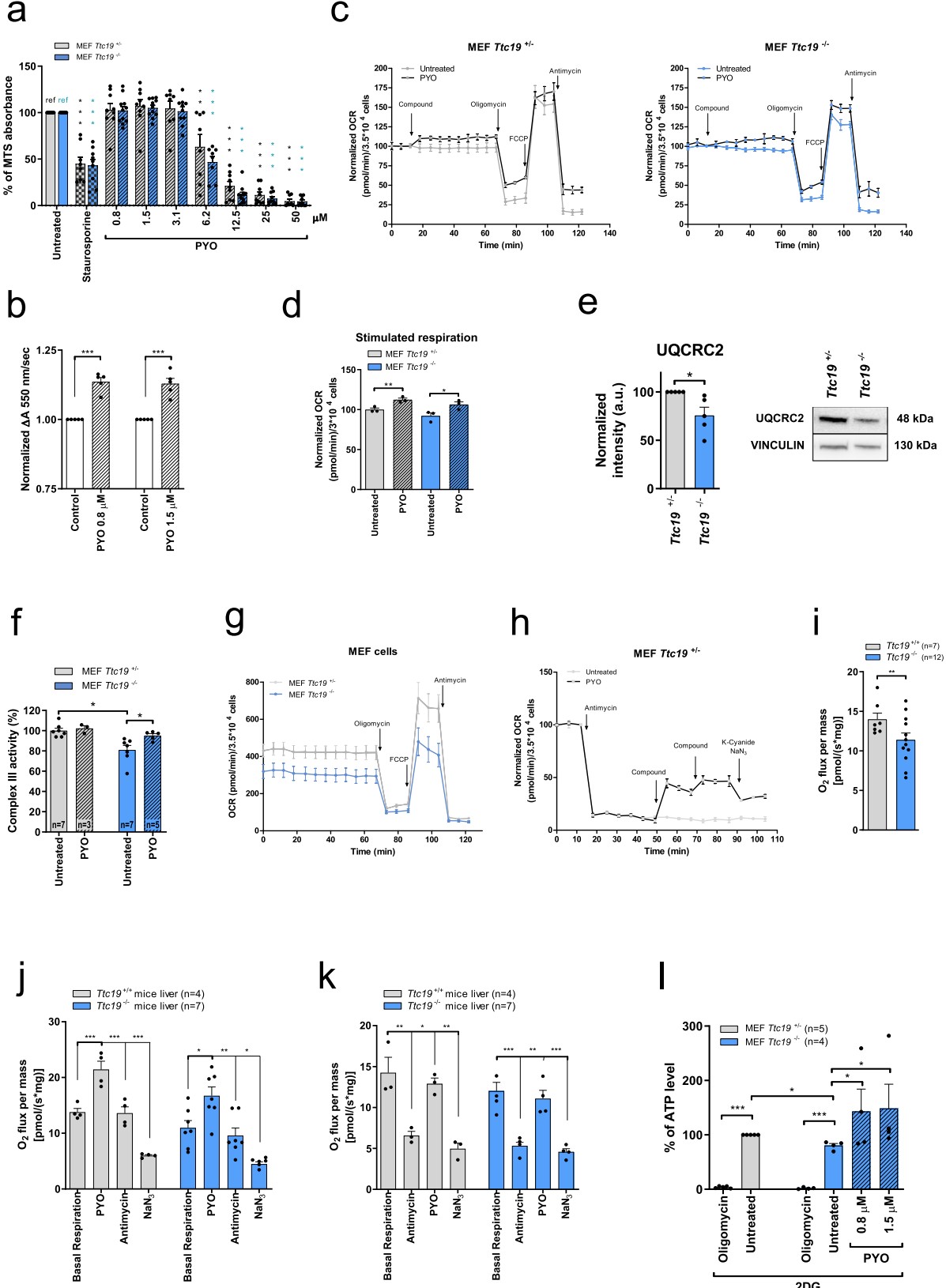

*Ttc19*$^{-/-}$ cell lines and found that their treatment with 3 μM or lower PYO concentrations for 24 h did not affect cell survival (Fig. 1a).

Next, we tested whether this nontoxic concentration of PYO is able to "replace" CIII function in isolated mitochondria, that is, to accept electrons from ubiquinol and donate them to cyt *c*, thus

restoring electron flow across the mitochondrial RC. We used alamethicin-permeabilized mouse liver mitochondria and assessed CIII function by measuring the reduction of cyt *c* in mitochondria in which CI, CIII, and CIV were inhibited by rotenone, antimycin A, and NaN₃, respectively. Upon addition of decylubiquinol and 0.8 or 1.5 μM PYO, cyt *c* reduction increased

**Fig. 1 Sublethal concentrations of PYO rescue ATP production and cell respiration in $Ttc19^{-/-}$ MEFs. a** Viability of $Ttc19^{+/-}$ and $Ttc19^{-/-}$ MEFs, assessed by MTS assay after 24 h with no (refs.) or increasing doses of PYO, using 4 μM staurosporine (ST) as control. Values are means of $A_{490}$ ± SEM ($n = 9$ assays). **b** Cyt $c$ reduction as $\Delta A_{550}$ in isolated mitochondria, with rotenone (2 μM), antimycin A (2 μg/ml), and NaN₃ (2.5 mM). PYO was added with 75 μM of reduced decylubiquinol. PYO-associated cyt $c$ reduction was normalized to that of untreated mitochondria ($n = 5$ measurements, three independent mitochondria preparations, mean values ± SEM). **c** Oxygen consumption rate (OCR) of $Ttc19^{+/-}$ and $Ttc19^{-/-}$ MEF was measured in three biological replicas with 1.5 μM PYO. Values (means ± SEM) were normalized against those recorded before PYO. **d** OCR was recorded after sequential addition of oligomycin (2 μg/ml); FCCP (600 nM); antimycin A (1 μM). Results are means ± SEM normalized to untreated sample ($n = 3$ independent assays). **e** Expression level of CIII subunit UQCRC2 by Western blotting (five biological replicas). Vinculin was used as a loading control. **f** CIII activities in mitochondria from $Ttc19^{+/-}$ and $Ttc19^{-/-}$ MEFs, before and after the addition of PYO, normalized to citrate synthase (means ± SEM). **g** OCR in the same cells measuring their basal respiratory capacity (means ± SEM, $n = 8$ independent assays). **h** PYO restores respiration of $Ttc19^{+/-}$ MEF after CIII inhibition by antimycin A. Values were normalized with respect to basal respiration recorded before antimycin A addition (considered as 100%) (means ± SEM; $n = 4$ biological replicas). **i–k** OCR levels in the liver from $Ttc19^{+/+}$ and $Ttc19^{-/-}$ mice. **i** shows basal respiration rate. PYO-dependent OCR was measured under basal condition. The sequence of additions was as shown, left to right. (means ± SEM). **l** Mitochondrial ATP content in $Ttc19^{+/-}$ and $Ttc19^{-/-}$ MEFs after treatment with PYO, using oligomycin as control. The same number of cells were grown in 5.5 mM glycolysis inhibitor 2-deoxyglucose (2-DG). Values are means of percentages of luciferase signals of treated vs untreated $Ttc19^{+/-}$ MEFs, ±SEM. One-way ANOVA with Dunnett's multiple comparison test or two-tailed Student's $t$ test (*$p < 0.05$; **$p < 0.01$; ***$p < 0.001$).

(as indicated by an increase in absorbance at 550 nm, associated with the chemical reduction of this mobile electron carrier), even when all complexes were inhibited (Fig. 1b). Accordingly, PYO was able to reduce cyt $c$ also in the absence of mitochondria, when decylubiquinol was present (Figure S1a). These results indicate that PYO is able to accept electrons from decylubiquinol and reduce cyt $c$, thus replacing the redox function of CIII. Figure S1b shows that PYO can accept electrons, at least in vitro, also from NADH, as expected[37]. However, in intact cells NADH depletion does not seem to occur, since the extracellular acidification rate reflecting glycolysis does not decrease (Figure S1c). Accordingly, 1.5 μM PYO caused a clear increase in basal (and stimulated) respiration rates in intact cells (Fig. 1c, d), as assessed by measuring the rate of mitochondrial oxygen consumption[38]. In agreement with the decreased amount of CIII subunits (Fig. 1e) and activity (Fig. 1f), respiration was lower in the homozygous $Ttc19^{-/-}$ MEF cells compared to the $Ttc19^{+/-}$ heterozygous cells used as controls (Fig. 1g).

To further prove that PYO was able to increase respiration even in the absence of functional CIII, we blocked CIII in $Ttc19^{+/-}$ MEFs with antimycin A and added PYO afterwards. Importantly, PYO-induced respiration was decreased by inhibitors of CIV (Fig. 1h). To assess whether PYO may exert a positive effect not only in cultured cells but also in critical diseased tissue, we isolated livers from wild-type (WT) and $Ttc19^{-/-}$ mice and assessed mitochondrial respiration by high-resolution oxygraphy. As expected, oxygen consumption, which was reduced in $Ttc19^{-/-}$ versus $Ttc19^{+/-}$ mitochondria (Fig. 1i)[21], was increased by PYO, under both basal conditions (Fig. 1j) and in the presence of the CIII inhibitor antimycin A (Fig. 1k). The increased oxygen consumption was sensitive to CIV blockers (Fig. 1j, k), confirming that PYO shuttled electrons to cyt $c$ rather than directly to oxygen.

A higher respiratory rate is expected to increase ATP synthesis[38], even though the correlation between respiration and ATP production is not linear (i.e., small changes in respiration lead to significant differences in ATP levels[39]). In order to measure the levels of ATP of mitochondrial origin, cells were cultured in the presence of 5.5 mM 2-deoxyglucose (2-DG) to eliminate the contribution of glycolysis to ATP synthesis. Low concentrations of PYO added to intact $Ttc19^{-/-}$ cells enhanced the ATP level (Figure S1d). Already at sub-μM levels, as low as 200 nM, PYO was able to increase mitochondrial ATP levels within 6 h. However, at concentrations >3 μM, this effect was abolished, likely because of PYO toxicity (Figure S1d). When data were normalized to the ATP level measured in $Ttc19^{+/-}$ MEFs, a decrease of ATP of mitochondrial origin occurred in $Ttc19^{-/-}$ cells (Fig. 1l), as expected based on the decreased CIII activity observed in various

tissues[21]; however, incubation of the TTC19$Ttc19^{-/-}$ cells with 0.8 or 1.5 μM PYO increased the levels of mitochondrial ATP by approximately two-fold.

PYO can donate electrons not only to cyt $c$ but also to molecular oxygen. Therefore, at high concentrations, PYO promotes the release of mitochondrial ROS, ATP depletion, and eventually cell death[32,36]. However, when applied at low concentration, PYO increased mitochondrial ROS production by only ~10% in both cell lines, a mild increase that is not sufficient to compromise cell survival (Fig. 2a). In addition, this ROS level failed to induce lipid peroxidation, either after 72 h of incubation (Fig. 2b and Figure S2a) or after treatment of the cells for 2 months (Fig. 2c and Figure S2b). Likewise, protein oxidation was not increased after incubation with PYO (Fig. 2d and Figure S2c for 72 h, and Fig. 2e for 2-month treatment). Accordingly, no upregulation of the antioxidant enzyme catalase occurred even after long-term incubation with PYO (Fig. 2f). At the same time, PYO stabilized the mitochondrial membrane potential and had a hyperpolarizing effect (Fig. 2g, h). Next, we investigated whether the hyperpolarizing effect of PYO required RC function. In the presence of rotenone (CI inhibitor), PYO was able to restore membrane potential (Figure S2d). We then pre-treated cells with rotenone, K⁺-cyanide (CIV inhibitor), and oligomycin (CV inhibitor), as well as atractyloside, an inhibitor of the adenine nucleotide transporter (ANT) that exports ATP from the mitochondrial matrix. Under these conditions, PYO was unable to rescue the decrease of the mitochondrial proton electrochemical membrane potential (essentially the electric gradient Δψm), caused by RC block. However, the activity of CIV (in the absence of cyanide) was sufficient to restore Δψm in the presence of both PYO and the other inhibitors, indicating that at the mechanistic level the beneficial action of PYO is indeed linked to respiration, downstream of CI and upstream of CIV (Fig. 2i and Figure S2e).

Since OXPHOS defects are often associated with ultrastructural alterations in cristae shape[40], we investigated whether mitochondrial morphology was altered in $Ttc19^{-/-}$ versus $Ttc19^{+/-}$ cells and whether PYO at the low concentrations had any effect on the structure of the organelle. Comparison of mitochondrial shape and size in transmission electron microscopy images in $Ttc19^{-/-}$ versus $Ttc19^{+/-}$ cells revealed that in naive KO cells mitochondria were more fragmented and their mean area was significantly lower (Fig. 3a). However, treatment with PYO for 72 h almost completely restored organelle size. Accordingly, expression of Mitofusin-2 (MFN2), a protein crucial to mitochondrial fusion[41], increased under PYO treatment (Fig. 3b). MFN2 also protects mitochondria against specific mitophagy via organellar

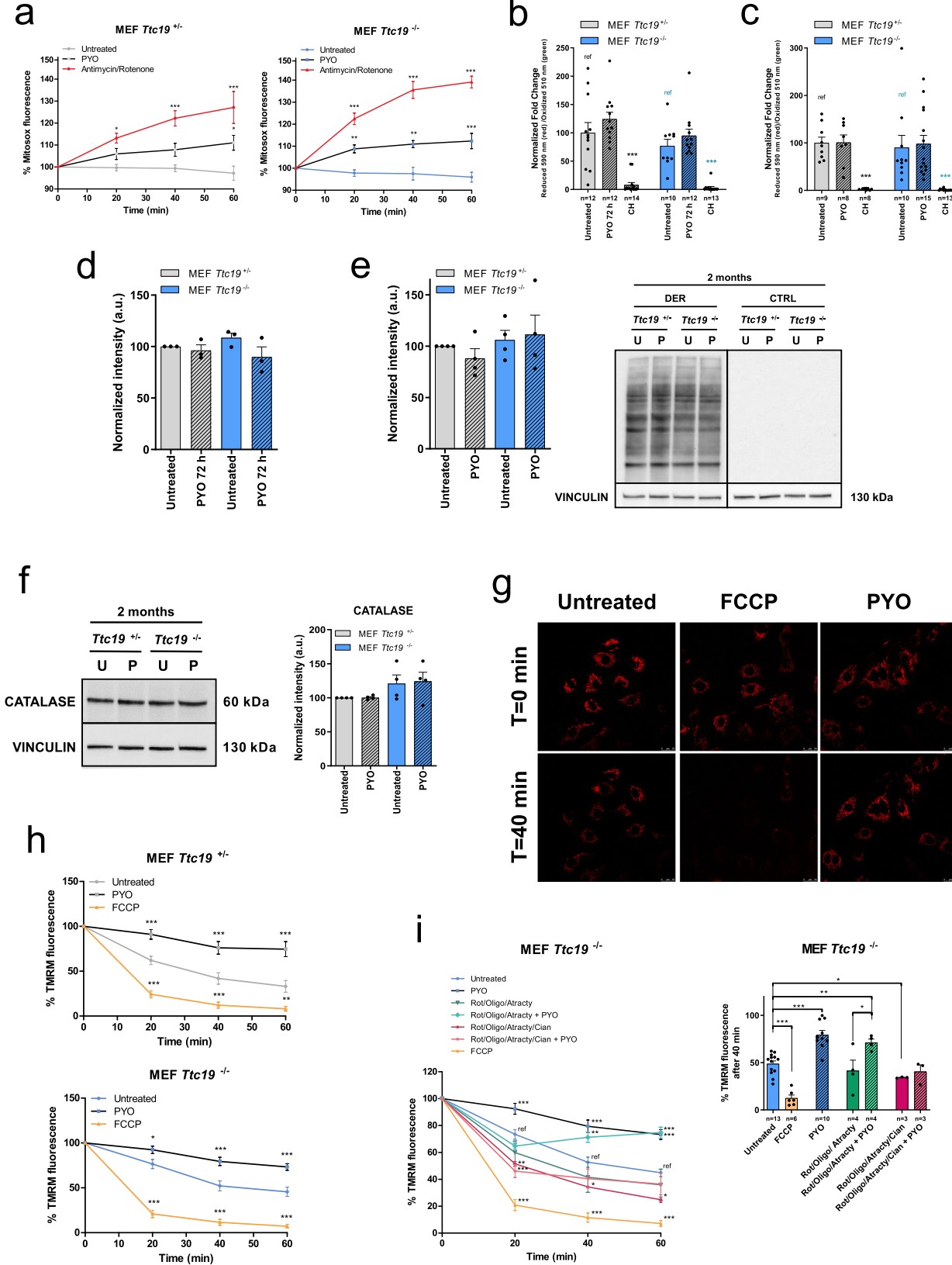

elongation. Thus, we measured the level of ATP5A (CV alpha subunit) and mitochondrial size, since enhancement of mitophagy should decrease both[42]. Indeed, $Ttc19^{-/-}$ cells were characterized by a lower ATP5A level and smaller mitochondria compared to controls (Fig. 3c), suggesting increased mitophagy and/or impaired biogenesis. PYO was able to significantly

increase also the level of ATP5A (Fig. 3c) with a tendency to enhance the level of proteins belonging to other RC complexes (Figure S2f). To investigate whether mitochondrial biogenesis was also increased by PYO, we investigated the levels of PGC-1α, the main transcription factor driving mitochondrial biogenesis[43]. The relative expression of $Pgc-1\alpha$ at both the transcript (Fig. 3d) and

**Fig. 2 Sublethal concentrations of PYO are not toxic in vitro. a** Mitochondrial ROS production by *Ttc19*[+/−] and *Ttc19*[−/−] MEFs measured by Mitosox fluorescence. Cells were either left untreated or treated with 1.5 μM PYO, and then fluorescence was monitored for 1 h. Antimycin A and rotenone were used as a positive control. Values are percentages against measurements before additions (means ± SEM, *n* = 4 independent replicas). **b, c** Quantification of lipid peroxidation in *Ttc19*[+/−] and *Ttc19*[−/−] MEFs untreated and treated with 1.5 μM PYO for 72 h (**b**) or for 2 months (**c**). Positive control: cumene hydroperoxide (CH). Data are means ± SEM. The number of the quantified images over three independent experiments is reported in the histogram . **d** Quantification of protein oxidation in *Ttc19*[+/−] and *Ttc19*[−/−] MEFs, both untreated and treated with 1.5 μM PYO for 72 h (means ± SEM, *n* = 3 independent experiments). **e** Representative oxyblot and quantification of protein oxidation level of *Ttc19*[+/−] and *Ttc19*[−/−] MEFs untreated and treated with 1.5 μM PYO for 2 months (means ± SEM, *n* = 4 independent experiments). **f** Representative Western blot and quantification of catalase expression in *Ttc19*[+/−] and *Ttc19*[−/−] MEF lysates untreated or treated with 1.5 μM PYO for 2 months (means ± SEM, *n* = 4 independent experiments). **g** Mitochondrial membrane potential of *Ttc19*[−/−] MEF determined by monitoring the intensity of TMRM confocal fluorescence. Scale bar: 25 μm. Positive control: FCCP-treated cells. Images are representative of two experiments. **h** Mitochondrial membrane potential of *Ttc19*[+/−] and *Ttc19*[−/−] MEFs by monitoring the intensity of TMRM fluorescence after 1.5 μM PYO. Values are percentages of the basal TMRM fluorescence recorded before treatments (means ± SEM, *n* = 10 independent experiments). **i** Membrane potential was measured as in (**h**), PYO was added 10 min after the mix of the indicated substances. The concentrations of cyanide, oligomycin, atractyloside, and rotenone were 1 mM, 0.5 μg/ml, 20 μM, and 1 μM, respectively. Data are percentages of the basal TMRM fluorescence recorded before treatments and are means ± SEM of independent experiments. Two-way ANOVA with Bonferroni post test, or one-way ANOVA with Dunnett's multiple comparison test, or two-tailed Student's *t* test were used (*$p < 0.05$; **$p < 0.01$; ***$p < 0.001$).

protein levels (Fig. 3e) was higher following PYO treatment, suggesting that PYO induced both mitochondrial fusion and biogenesis. Accordingly, the protein levels of TOM20, a component of the protein import machinery (Fig. 3f), and of some RC complex subunits (Figure S2f) tended to be slightly increased under PYO treatment.

We have recently reported that CIII-deficient cells are characterized by reduced Wnt signalling. Deficit in mitochondrial ATP synthesis decreases Wnt signalling by impairing the function of the endoplasmic reticulum $Ca^{2+}$-ATPase SERCA, thereby causing reduced endoplasmic reticulum (ER) calcium concentration and ER stress. In turn, ER stress decreases Wnt signalling via the transcription factor CHOP, which functions as a specific inhibitor of Wnt/T cell factor (TCF) signalling[20]. Contrariwise, increased ATP concentration in CIII-deficient human fibroblasts via exposure to phosphoenolpyruvate restores Wnt signalling. Therefore, we tested by Western blotting and in vivo in a Wnt-reporter zebrafish line whether PYO, shown above to enhance ATP level, was also able to increase Wnt signalling. Indeed, PYO, used at low concentrations, rescued Wnt signalling reduction in CIII-deficient MEFs (Fig. 3G) and in vivo, using the well-established *Tg(7xTCFX.lasiam:EGFP)ia4* zebrafish reporter line for canonical Wnt signalling[20,44] (Fig. 3h).

**PYO increases respiration and ATP in human *TTC19*-mutated cells.** Next, we tested the effect of PYO on human fibroblasts derived from three different patients harboring *TTC19* mutation (see Table 1). Similar to MEFs, both healthy and TTC19-truncated human fibroblasts exposed to PYO at sublethal concentrations (Fig. 4a) increased ATP content (Fig. 4b and Figure S3a). When data were normalized to the ATP level measured in human healthy fibroblasts, reduced ATP content could be observed in patients (Fig. 4c), as expected, whereas incubation of cells with 0.4 or 0.8 μM PYO restored the mitochondrial ATP content, to the same levels observed in healthy fibroblasts. Sublethal concentrations of PYO were associated with slightly higher basal respiration in healthy as well as in *TTC19*[−/−] MEFs (Fig. 4d, e and Figure S3b). To further prove that PYO was able to increase respiration even in the absence of functional CIII, we blocked CIII in healthy fibroblasts with antimycin A and added PYO afterwards. As illustrated in Fig. 4f, PYO increased respiration also under these conditions. Similar to MEFs, also in human fibroblasts PYO stabilized the mitochondrial membrane potential (Fig. 4g and Figure S3c).

Mitochondria from mutant human fibroblasts showed a fragmented network in comparison to healthy fibroblasts, but PYO was able to restore the elongated phenotype (Fig. 5a) and

enhance TOM20 protein level (Figure S3d). Again, PYO triggered a mild ROS release (Fig. 5b and Figure S3e). However, this was not sufficient to oxidize proteins (Fig. 5c) or lipids (Fig. 5d and S3f) or to promote catalase upregulation (Fig. 5e).

To further explore if PYO might exert a beneficial effect in CIII diseases linked to the mutations of other important factors, fibroblasts from a patient with a homozygous *LYRM7* mutation as well as those from a *BCS1L*-mutated patient (see Table 1) were assessed for the same parameters. All the results were comparable to those observed in *TTC19*-mutated patients, demonstrating that PYO at sublethal concentrations triggers the same response, leading to a significant improvement in mitochondrial fitness (Figures S4 and S5).

**PYO rescues motor deficit in *dTTC19* KO *Drosophila*.** These promising in vitro results prompted us to test PYO on locomotor activity in vivo. First, we used a mutant *Drosophila melanogaster* TTC19-null fly model in which we had previously observed reduced activity of CIII and strong bang sensitivity (i.e., reduced neuromotor activity)[11]. The *dTTC19* KO adult flies obtained here by CRISPR/Cas9 also showed strong bang sensitivity and markedly reduced production of mitochondrial ATP after 24 h fasting (Fig. 6a). Before applying the redox cycler PYO, its toxicity was assessed in WT flies (*w*[1118]) to establish the suitable range of nontoxic dosages in Drosophila. The injection of <50 pmol PYO/fly into the hemolymph of flies was shown not to be toxic. (Fig. 6b). A strong increase in ATP content (Fig. 6c) and reduced bang sensitivity (i.e., enhanced locomotor activity) were observed in the *dTTC19* KO 5-day-old flies treated with only 1.0 pmol of PYO/fly compared to controls (*w*[1118]; *w*[1118] injection control flies, and *dTTC19* KO injection control flies) (Fig. 6d). PYO was able to partially recover the activity of almost completely paralyzed 12-day-old flies as well (Fig. 6e). In addition, when the percentage of flies recovering from the bang test was compared between WT injection control flies and KO injection control flies, differences were significant for all bang test heights both in young (2.8 cm: $\chi^2$ = 18, $p < 0.0001$; 5.6 cm: $\chi^2$ = 11, $p < 0.001$; 8.4 cm: $\chi^2$ = 5, $p$ = 0.03; 11.2 cm: $\chi^2$ = 6, $p = 0.01$) (Fig. 6d) and old flies (2.8 cm: $\chi^2$ = 50, $p < 0.0001$; 5.6 cm: $\chi^2$ = 43, $p < 0.0001$; 8.4 cm: $\chi^2$ = 32, $p < 0.0001$; 11.2 cm: $\chi^2$ = 13, $p < 0.001$) (Fig. 6e). In the latter, differences were more prominent. Finally, when the percentage of flies recovering from the bang test was compared between KO injection control flies and KO treated with PYO (*dTTC19* KO + PYO), differences were significant for all bang test heights both in young (2.8 cm: $\chi^2$ = 6, $p = 0.01$; 5.6 cm: $\chi^2$ = 8, $p < 0.01$; 8.4 cm: $\chi^2$ = 10, $p < 0.01$; 11.2 cm: $\chi^2$ = 23, $p < 0.0001$)(Fig. 6d) and old flies (2.8 cm: $\chi^2$ = 14, $p < 0.001$; 5.6 cm: $\chi^2$ = 11, $p < 0.001$; 8.4 cm:

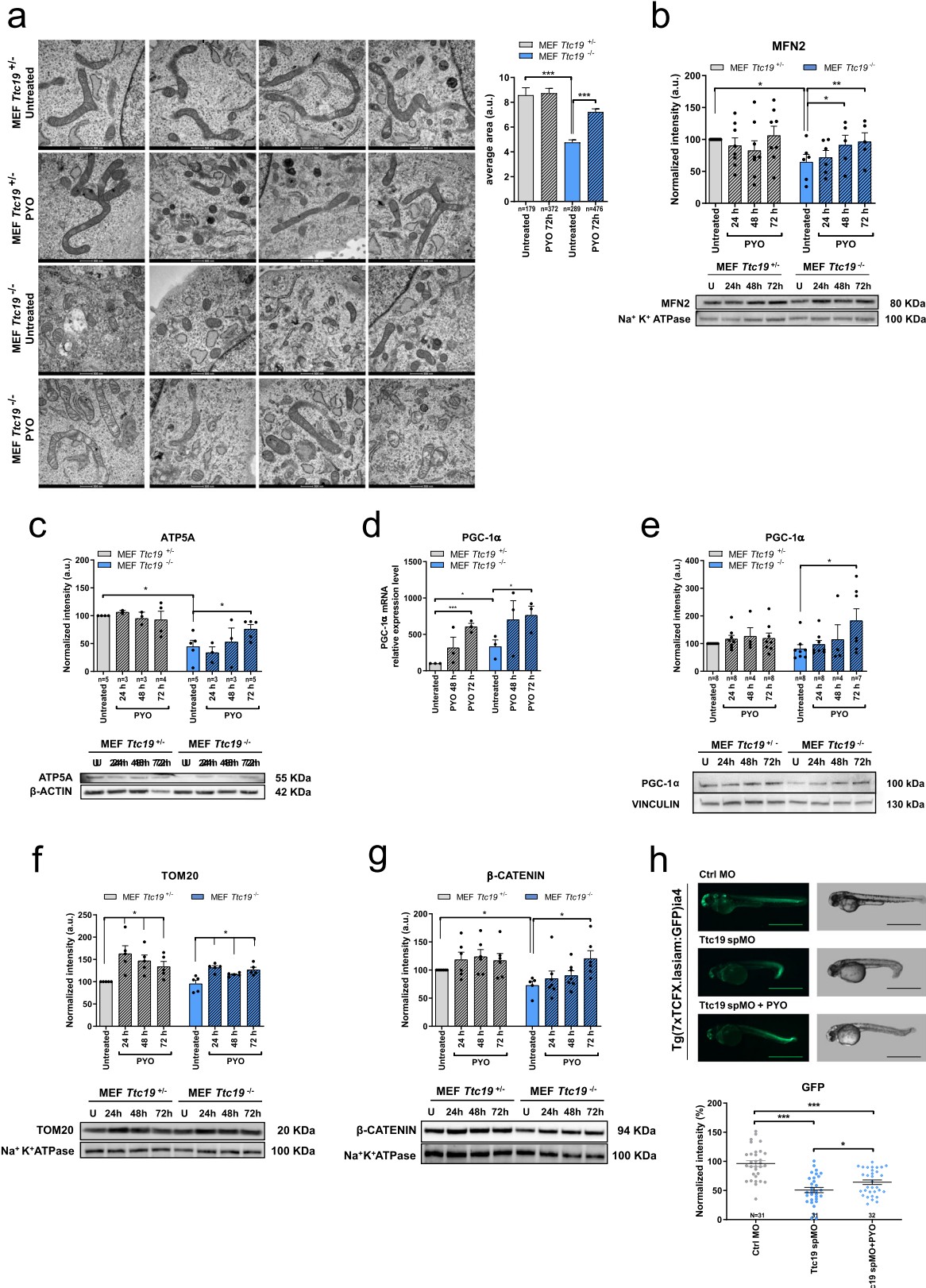

$\chi^2 = 12$, $p < 0.001$; 11.2 cm: $\chi^2 = 13$, $p < 0.001$) (Fig. 6e). Again, differences were more prominent in old flies (Fig. 6d, e). Of interest, young KO flies seemed to exhibit more difficulties but also to benefit more from treatment with PYO when the test was most difficult, that is, at the 11.2 cm height (Fig. 6d). Thus, PYO triggers increased ATP level in $dTTC19$ KO flies and allows

recovery of biochemical and behavioral phenotypes similar to those of WT flies. Interestingly, a two-fold increase in ATP content after 1.0 pmol of PYO treatment was observed also in WT flies (Fig. 6c). The lack of toxicity was confirmed also in vivo in this model, as protein oxidation did not occur (Fig. 6f), and two injections of PYO did not significantly decrease vitality within

**Fig. 3 PYO improves mitochondrial morphology in *Ttc19* KO mouse embryonic fibroblasts. a** Representative transmission electron microscopy images of MEF *Ttc19*$^{+/-}$ and MEF *Ttc19*$^{-/-}$ left untreated or treated for 72 h with 1.5 µM PYO. The average mitochondrial cross-sectional area was calculated for each condition using the ImageJ software. Quantification is shown (means ± SEM, *n* of the counted mitochondria is reported in the image). Scale bars: 500 nm. **b, c** Protein levels of Mitofusin-2 (**b**, *n* = 6 independent experiments), ATP5A (**c**, the value of *n* is reported in the figure), were assessed by Western blot. MEF *Ttc19*$^{+/-}$ and MEF *Ttc19*$^{-/-}$ were left untreated or treated for 24, 48, or 72 h with 1.5 µM PYO. Quantification by densitometry (means ± SEM) from independent experiments is reported in the upper part of the panels, while representative blots are presented in the lower parts. Na$^+$K$^+$ ATPase or β-actin were used as loading controls. **d** Real-time PCR analysis of the transcription of *Pgc-1α* was assessed in MEF 9*Ttc19*$^{+/-}$ and MEF *Ttc19*$^{-/-}$ either left untreated or treated with 1.5 µM PYO for 48 or 72 h. All values were normalized on β-actin expression. The values are reported as means of percentages of RNA expression ± SEM (*n* = 3 independent replicas) with reference to the untreated *Ttc19*$^{+/-}$ sample, as indicated in the figure. **e, f** as in (**b, c**) using antibodies against PGC-1α (**e**, value of *n* is reported in the image) and TOM20 (**f**, *n* = 5 technical replicates over three biologically independent samples). Vinculin or Na$^+$K$^+$ ATPase were used as loading controls. Data are shown as means ± SEM. **g** One hundred nanomoles of PYO enhances Wnt signaling in MEF *Ttc19*$^{+/-}$ and MEF *Ttc19*$^{-/-}$ cells incubated with PYO for the indicated times (means ± SEM, *n* = 7 technical replicates over three biological independent samples). **h** PYO significantly increases Wnt signaling in a zebrafish reporter line following incubation of the embryos with nonlethal 100 nM concentration for 24 h (*n* = 31 embryos for each condition). Scale bars: 1 mm. Statistical significance (one-way ANOVA with Dunnett's multiple comparison test or two-tailed Student's *t* test) was calculated for all panels (*$p < 0.05$; **$p < 0.01$; ***$p < 0.001$).

**Table 1 Genetic mutations in patients affected by complex III disease.**

| Patient | Mutated gene | Reference paper | Gene mutation | Effect |
|---|---|---|---|---|
| #1 | TTC19 | Ghezzi et al., 2011 | Homoz nucleotide change c.656T > G (p.Leu219X) | Truncated protein |
| #2 | TTC19 | Ghezzi et al., 2011 | Homoz nucleotide change c.656T > G (p.Leu219X) | Truncated protein |
| #3 | TTC19 | Ardissone et al., 2015 | Homoz rearrangement c.782_786delinsGAAAAG (p. Glu261Glyfs*8) | Frameshift with a predicted premature termination |
| #4 | LYRM7 | Dallabona et al., 2016 | homoz c.193_195dup (p.L66dup) | |
| #5 | BCS1L | Fernandez-Vizarra et al., 2007 | Heteroz nucleotide change c. 217C > T (p. R73C) heteroz nucleotide change c. 1102T > A (p. F368I) | |

3 days after the second injection, with survival rates ranging from 95 to 97% (Fig. 6h). In agreement with our results on MEF cells, PGC-1α levels increased after PYO treatment also in this model (Fig. 6g).

**PYO ameliorates *TTC19* KD zebrafish phenotype**. Next, we decided to study the effect of PYO in the vertebrate *Danio rerio* model for CIII disease. By using morpholino-mediated transient KD of the *TTC19* $^{mRNA20}$ (Ttc19spMO), we observed reduced respiration (Fig. 7a, b). When we added 100 nM PYO, which is nontoxic to fish (Fig. 7c), into the fish-surrounding water starting from 6 h post fertilization (hpf) and replaced it twice a day until 72 h.p.f., mitochondrial respiration increased significantly (Fig. 7a, b). Similar to *Drosophila*, also zebrafish was able to recover, albeit only partially, the ability to move. In particular, the evoked touch response (ETR) (i.e., the ability of the embryos to swim following mechanical stimulation) was almost completely absent in *TTC19* KD animals. Some of these individuals completely recovered ETR after 100 nM PYO treatment, and an increase in ETR was observed in most of the remaining treated individuals (Fig. 7d, e and see Movie 1). In addition, the sagittal axis shortening, tail and cranio-facial deformity, and lower pigmentation observed in the *TTC19* KD animals was also ameliorated in PYO-treated fish (Fig. 7f). Since Ttc19 spMO fish show an altered morphology compatible with impaired functionality of motoneurons (MN) and neuro-muscular junction (NMJ), we analyzed nmx1[45] expression, MN differentiation, and the ratio MN/Area (Figure S6a). The peripheral nervous system was investigated in *Tg(mnx1:mGFP)*, which specifically highlights MNs in vivo. *TTC19* morphants showed reduced mnx1 expression and a general defect in MN development in comparison to control morphants. PYO treatment did not improve mnx1 expression or enhanced the ratio of the number of MNs/area, but favored MN axon elongation, suggesting partial recovery of MNJ.

Since Wnt signalling has previously been linked to NMJ formation[46], Wnt signalling recovery may partially explain axon elongation.

**Lack of toxicity in mice upon long-term PYO treatment in vivo**. Given the promising results obtained in both *Drosophila* and zebrafish, with the perspective of using PYO also in mammals, we assessed whether the application of PYO for longer times may cause toxicity related to oxidative stress. Therefore, we treated adult mice with intraperitoneal injections of PYO (10 nmol/gbw) once a day, 5 days a week, for 2 months (*n* = 8). There was no arrest in growth (Fig. 8a) or sign of malaise (see Movie 2) and even the long-term treatment did not cause an inflammatory phenotype (Fig. 8b) in the animals. In addition, different tissues were examined by histology (hematoxylin–eosin staining) showing no gross alterations (Fig. 8c), and the ultrastructure of nucleus and mitochondria in the liver (Fig. 8d), as well as in the brain and muscle (Figure S6b), of these WT mice was not affected. Furthermore, protein oxidation level (Fig. 8e) as well as CAT expression levels (Fig. 8f) were not significantly altered in liver, brain, and heart of animals treated with PYO. To verify that PYO is able to reach target organs after intraperitoneal (i.p.) administration, we measured its levels in the liver over a 4-h period after a single injection. PYO was present at concentrations <1 nmol/gbw (*n* = 3 biologically independent samples).

**Discussion**
Here, we tested the properties of the bacterial redox cycle molecule PYO, as a potential treatment to alleviate the defects associated with CIII disease. As previously mentioned, PYO can accept electrons from both nicotinamide adenine dinucleotide (NADH) and ubiquinol; its measured electrochemical redox potential is indeed closer to that of the ubiquinone/ubiquinol couple. PYO can donate electrons, predominantly to cyt *c*, but

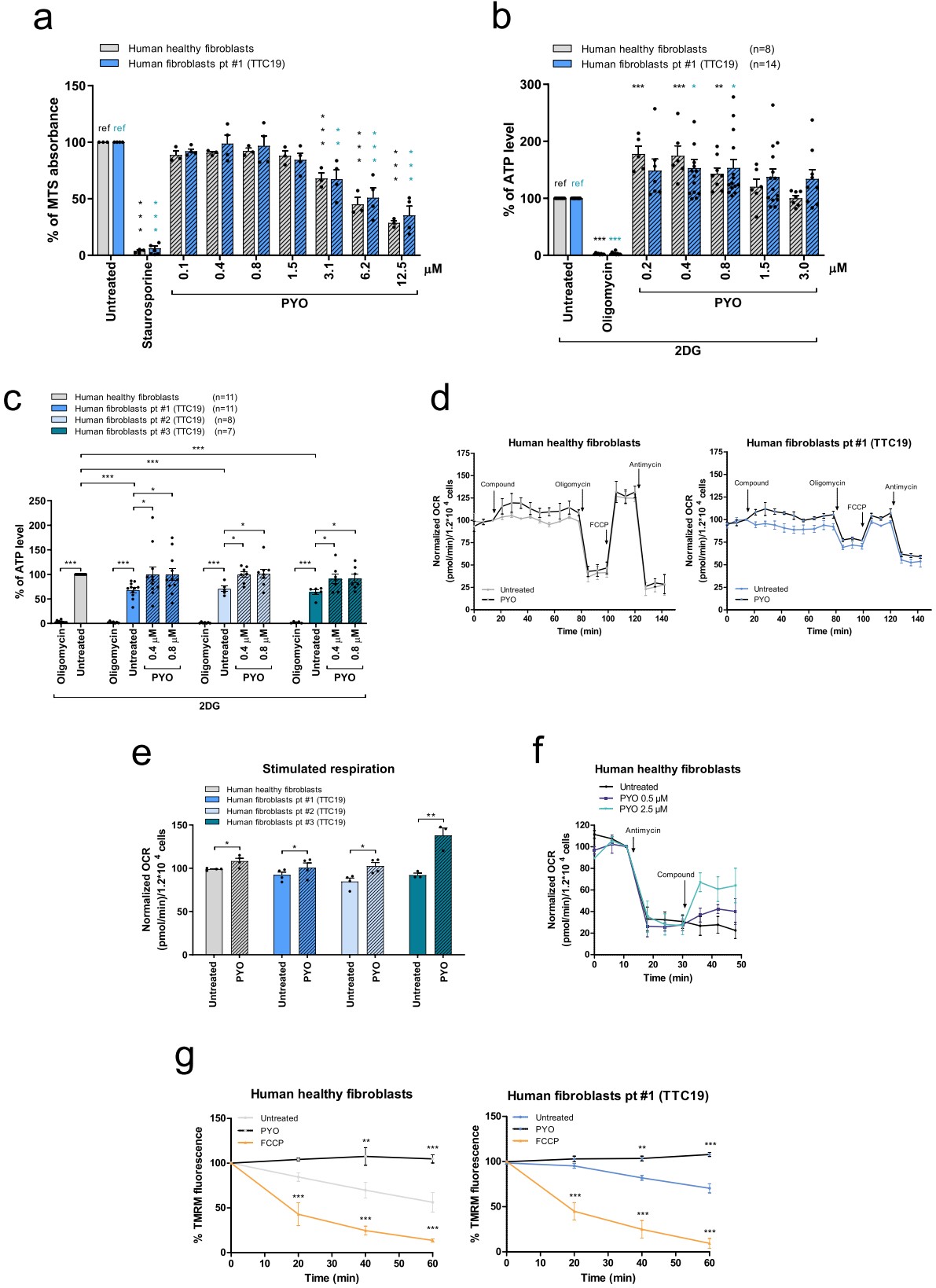

also to molecular oxygen, accounting for a strong release of mitochondrial ROS and toxicity when used at high concentrations[36]. However, as illustrated in this paper, the positive effect of PYO used at very low concentrations (sub-μM range) on respiration clearly prevailed on the ROS-inducing effect and allowed an up to three-fold increase in ATP levels in different systems examined here. PYO was indeed able to improve bioenergetics in fibroblasts derived from five patients with

**Fig. 4 PYO increases respiration and ATP production, and recovers mitochondrial morphology in human fibroblasts from patients with _TTC19_ gene mutations. a** MTS assays were performed on human healthy fibroblasts ($n = 3$ independent assays) and fibroblasts from a patient harboring a homozygous pathogenic mutation in the _TTC19_ gene (fibroblasts pt#1) ($n = 4$ independent assays). Cells were either left untreated or treated with different dosages of PYO for 24 h to determine the highest concentration of PYO that did not affect cell survival. Positive control: 4 µM staurosporine. Data are percentages (means ± SEM) of MTS absorbance at 490 nm vs untreated sample (ref.). **b**, **c** Mitochondrial ATP content in human healthy fibroblasts and patients' fibroblasts after treatment with PYO. Oligomycin was used as a control. Cells were treated for 1 h in a fresh medium, in which glucose had been replaced with 5.5 mM 2-DG to inhibit glycolysis. In **b**, values are reported as the percentage of luciferase signal vs. untreated sample (ref). In **c**, values are percentage of luciferase signal with reference to the untreated human healthy fibroblasts (in gray). Values are means ± SEM of independent experiments. **d**, **e** OCR was measured in human healthy fibroblasts and patient #1 fibroblasts (**d**) in the presence or absence of 0.8 µM PYO. Values were normalized to basal respiration before PYO (means ± SEM, $n = 4$ independent experiments). Bioenergetic parameters were calculated after sequential addition of oligomycin (2 µg/ml); FCCP (600 nM); antimycin A (1 µM). Stimulated respiration is reported in (**e**). Values are means ± SEM against untreated sample ($n = 4$ independent experiments). **f** OCR was measured in human healthy fibroblasts to assess the capability of PYO to restore cell respiration after antimycin A inhibition of CIII. Values were normalized vs. basal respiration recorded before antimycin A (means ± SEM, $n = 3$ independent experiments). **g** Mitochondrial membrane potential of human healthy fibroblasts and patient #1 fibroblasts was assessed monitoring the intensity of TMRM fluorescence after 0.8 µM PYO addition as in Fig. 2h (means ± SEM, $n = 4$ independent experiments). Two-way ANOVA with Bonferroni post test, or one-way ANOVA with Dunnett's multiple comparison test, or two-tailed Student's _t_ test were determined (*$p < 0.05$; **$p < 0.01$; ***$p < 0.001$).

dysfunction in three different factors involved in assembly and/or stability of CIII, namely TTC19, BCS1L, and LYRM7. Besides these cellular systems, a benefit was also observed in whole organisms such as _Drosophila_ and zebrafish.

The question arises whether the mechanism responsible for this effect is similar to that proposed for AOX. While this is reasonable, we observed that Δψm could be maintained even in the presence of the CI inhibitor rotenone. Therefore, an additional effect of PYO may also exist, since PYO is able to accept electrons from NADH[32,34]. An important observation is that PYO enhanced respiration also in the presence of CIII inhibition by antimycin A and had a stabilizing effect on the membrane potential. This effect was still observable when CI, the ATP synthase, and ANT were pharmacologically blocked, but PYO was unable to maintain Δψm upon simultaneous inhibition of CIV. Altogether, these data suggest that PYO alleviates energy decrease due to CIII dysfunction by a mechanism that requires the function of RC CIV, while the function of CI is dispensable.

PYO exerts a beneficial effect on mitochondrial morphology, which is intimately linked to the bioenergetic efficiency of these organelles[47]. Upon overexpression of MFN2, which is crucially involved in mitochondrial fusion, mitochondrial metabolism is activated. In particular, glucose oxidation, the Krebs cycle, and proton leak are enhanced, along with an increase in mitochondrial membrane potential and RC biogenesis[48]. We observed both an instantaneous increase in respiration by PYO and a longer-term effect after a 3-day treatment, indicating that low, non-cytotoxic concentrations of PYO can help to maintain mitochondrial homeostasis, possibly through its ability to trigger mild ROS production. Mild oxidative stress can elicit increased mitochondrial fusion and biogenesis in order to re-establish the respiration rate and ATP production, thus controlling metabolic adaptation[49]. Accordingly, overexpression of AOX, which prevents ROS production, worsens the myopathic phenotype of a muscle-specific _Cox15_$^{-/-}$ mouse model[26]. In contrast, highly fragmented mitochondria are a sign of damage, being usually eliminated through mitophagy, to restore mitochondrial homeostasis[50]. Recovery of mitochondrial function by PYO is indicated by the following evidence: (i) stabilization of membrane potential, (ii) increase in Mfn2 levels and mitochondrial fusion, and (iii) increase in the expression levels of PGC-1α and of subunits of the RC complexes. Altogether, these data fit with previously reported data showing the beneficial effects of increasing mitochondrial biogenesis in mouse models of mitochondrial disease[27–29]. However, given that PYO has an instantaneous effect on mitochondrial membrane potential and respiration, whereas the effect on organelle morphology appears

72 h after treatment, PYO is unlikely to promote ATP level increase (measured 6 h after treatment) uniquely as a secondary effect of mitochondrial biogenesis and mitodynamics. A further possibility to enhance mitochondrial biogenesis is via Wnt signaling, as this pathway can increase PGC-1α levels[51,52]. Indeed, PYO was able to enhance Wnt signaling both in vitro and in vivo, in agreement with our previous study where ATP supplementation was able to rescue reduced Wnt signaling in fibroblasts from patients with TTC19 dysfunction[20].

Taken together, our data show that ATP levels increase after treatment with low, sublethal concentrations of PYO, which do not cause strong ROS production. Thus, within a finely tuned concentration range, PYO is capable of increasing OXPHOS-produced ATP levels without causing excess oxidative stress. We propose that PYO efficiently recovers mitochondrial function by shuttling electrons from NADH and/or ubiquinol to CIII and thus acts directly on ATP production, therefore ameliorating bioenergetic efficiency. This indicates that sub-µM concentrations of PYO are a promising therapeutic strategy against CIII disorders. Importantly, long-term treatment with low concentrations of PYO in mice did not cause toxicity or oxidative stress, in agreement with previous studies showing that even 10 µM i.p. injected PYO in BALB/c mice significantly increased glutathione-S-transferase activity only in the heart, but not in other tissues[53]. Although detection of PYO and of its metabolites is not an easy task[54], our pharmacokinetic profiling was able to reliably detect PYO, and indicated that optimization of PYO administration in mice is required before testing its effect on behavioral parameters in vivo.

## Methods

**Reagents**. PYO, oligomycin, staurosporine, rotenone, antimycin A, NaN₃, and FCCP were obtained from Sigma-Aldrich. All compounds were dissolved in dimethyl sulfoxide (DMSO) (Sigma-Aldrich). For _D. melanogaster_ experiments PYO was dissolved in EtOH. The concentration of PYO was controlled for each batch by high-performance liquid chromatography (HPLC). Cells or zebrafish treated with 0.2% DMSO or flies with EtOH were used as a negative control.

**Cell cultures**. MEF _Ttc19_$^{+/-}$ and MEF _Ttc19_$^{-/-}$ cells were cultured in Dulbecco's modified Eagle's medium (DMEM, Invitrogen), supplemented with 10% fetal bovine serum (FBS, BioSpa), 10 mM HEPES (Life Technologies), 100 U/ml penicillin and 100 U/ml streptomycin (Life Technologies), and 1× non-essential amino acids (Life Technologies). Mouse lines were obtained by the authors and described previously[21]. MEFs were derived in-house from 12.5 d.p.c. embryos according to standard protocols.

Human immortalized fibroblasts were obtained from healthy human subjects and patients with deficiency of the RC CIII, due to different mutations in _TTC19_ or _LYRM7_ or _BCS1L_ genes (see Table 1)[11]. These human cells were obtained from the Biobank at Besta Institute, Milan and cultured in DMEM supplemented as stated

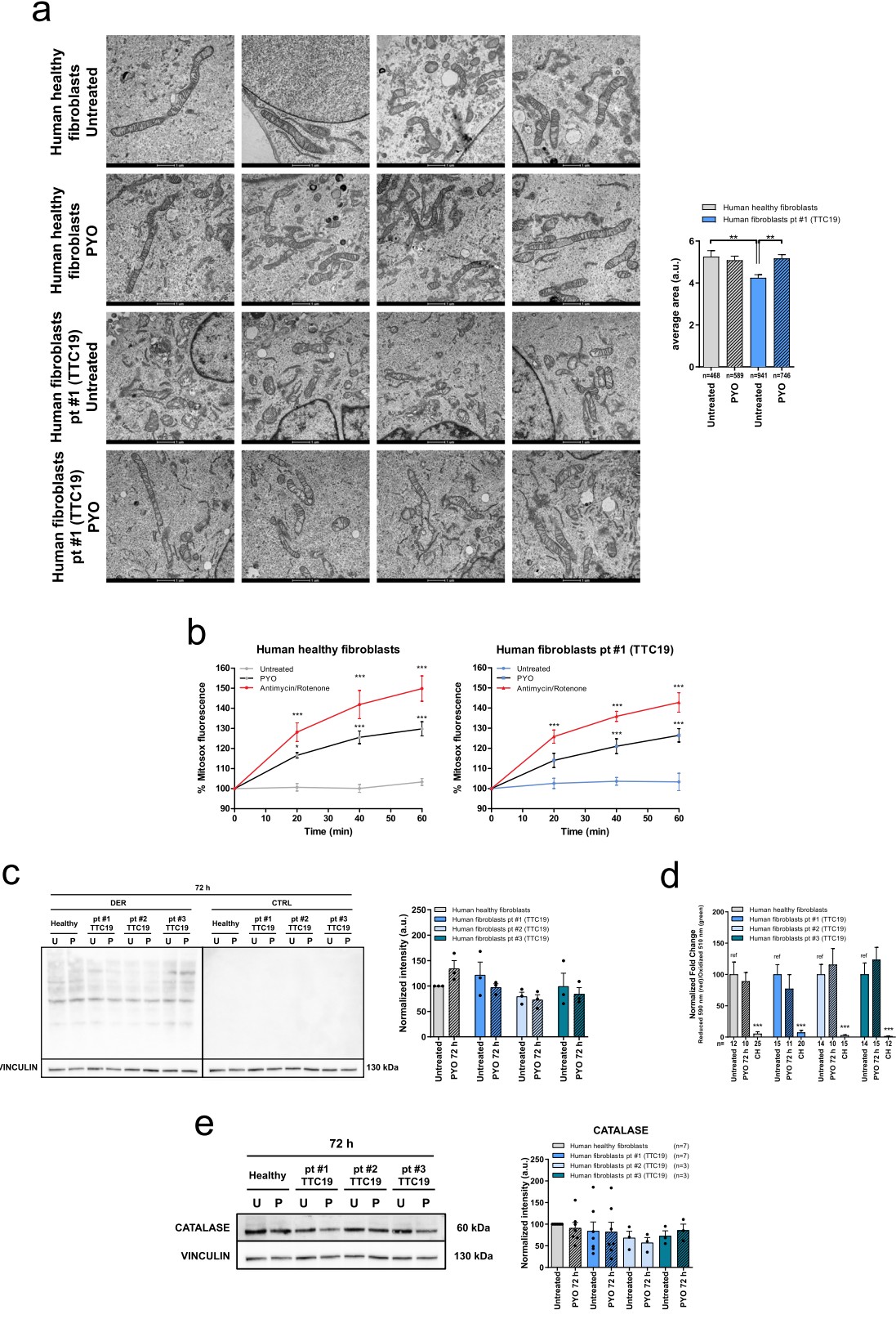

above. All cells were maintained at 37 °C and 5% $CO_2$. All cell lines used tested negative for mycoplasma contamination.

Informed consent for participation in this study was obtained from all investigated subjects in agreement with the Declaration of Helsinki. The experiments were approved by the ethical committee of the IRCCS Foundation Neurological Institute "C. Besta", Milan, Italy.

**Fly stocks and maintenance**. Flies were reared at 23 °C, 70% relative humidity, under 12-h light/12-h dark cycles (LD 12:12) on cornmeal standard food. For the starvation experiment, flies were maintained on 1% agar. d*TTC19* KO mutant line was generated by deleting the entire CDS of the CG15173 gene using the CRISPR/Cas9 system (WellGenetics, Taipei City, Taiwan). The $w^{1118}$ injection strain was used as a control. Five-day-old KO male flies and/or 12-day-old KO male flies were used in the study.

**Fig. 5 PYO recovers mitochondrial morphology in human fibroblasts from patients with *TTC19* gene mutations, with no oxidation toxicity.**
**a** Representative transmission electron microscopy images of human healthy fibroblasts and fibroblasts from patient #1, untreated or treated for 72 h with 0.8 μM PYO. Sample preparation and images analysis were performed as in Fig. 3a. Quantification is shown (means ± SEM) and the number of mitochondria counted is reported in the histogram. Scale bar: 1 μm. **b** Mitochondrial ROS production by human healthy and by fibroblasts from patient #1 was determined by measuring the intensity of Mitosox fluorescence as in Fig. 2a. PYO was added at a final concentration of 0.8 μM. Values are means ± SEM ($n = 4$ independent experiments). **c** Representative oxyblot and the relative quantification of the protein oxidation level of healthy fibroblasts and patients' samples, both untreated and treated with 0.8 μM PYO for 72 h (means ± SEM, $n = 3$ independent experiments). **d** Quantification of the lipid peroxidation in human healthy fibroblasts and fibroblasts from patients #1, #2, #3, untreated or treated for 72 h with 0.8 μM PYO. Cumene hydroperoxide (CH) was used as a positive control. Representative images are presented in Figure S3f (means ± SEM). The number of images quantified, recorded in three independent experiments, is reported. **e** Representative Western blot and the relative quantification of the catalase expression level in human healthy fibroblasts and fibroblasts from patients carrying *TTC19* mutations, both untreated and treated with 0.8 μM PYO for 72 h (means ± SEM of independent experiments). Statistical significance (two-way ANOVA with Bonferroni post test or two-tailed Student's $t$ test) was determined for all panels (*$p < 0.05$; **$p < 0.01$; ***$p < 0.001$).

**Zebrafish lines and morpholino models.** Zebrafish (animals and embryos) were maintained according to standard rules and conditions. All procedures were conducted according to the guidelines of the Local Ethical Committee at the University of Padua and of the National Agency, and with the supervision of the Central Veterinary Service of the University of Padova (in compliance with Italian Law DL 116/92 and further modifications, embodying UE directive 86/609).

WT zebrafish were of the Tübingen (Tü) strain and are currently stabled at the zebrafish facility of Padua University. Zebrafish embryos were analyzed using a Leica M165FC microscope. All images were acquired with a Nikon DS-F12 digital camera. All images were acquired with the same exposure parameters and processed with GIMP 2.0.

Custom Morpholinos were purchased from Genetools LLC. Ttc19 spMO (5′-AAGGCTGATGTGAAAGCAAATCTGA-3′) affects the exon 2 splicing acceptor site in *D. rerio* ttc19 messenger RNA (mRNA), standard Ctrl Morpholino (Ctrl MO) is composed by random sequence oligos, which does not significantly affect mRNA processing. Microinjection is performed on randomly separated sibling embryos at one-cell stage, adding 30 ng/embryos of morpholino. The compound was added between 8 and 8.5 hpf, immediately after chorion punching with a thin needle. Chorions were manually excided at 24 hpf and images were acquired at 36 hpf.

**Animal care and handling.** Animal experiments and care complied with the institutional guidelines and were approved by Italian authorities (both the local Ethics Committee (OPBA, Organismo preposto al benessere animale) at the University of Padova and the Italian Ministry for Health (474/2020-PR)). Experiments were carried out with the supervision of the Central Veterinary Service of the University of Padova (in compliance with Italian Law DL 116/92, embodying UE directive 86/609). Six-week-old C57BL6/J mice were maintained on a standard rodent chow diet with 12 h light/dark cycles, at room temperature of 19–22 °C, with 40/60% humidity. Mice received 10 nmol/g of PYO by i.p. injection 5 days a week for 9 weeks. Body weight was recorded weekly. At the end of the treatment, blood was drawn from the facial vein into EDTA-coated tubes and the animals were sacrificed. For histological analysis, tissues were fixed overnight in 4% paraformaldehyde in phosphate-buffered saline (PBS), and then dehydrated through exposure to progressively higher concentrations of ethanol and embedded in paraffin.

**Isolation of mitochondria and cyt *c* reduction assay.** Mouse liver was immediately immersed in an ice-cold isolation medium (250 mM sucrose, 5 mM HEPES, 2 mM EGTA; pH 7.5). The liver tissue was homogenized in the same solution and mitochondria were then isolated by conventional differential centrifugation, as previously described[55]. The protein content was determined by the Bradford protein assay, with bovine serum albumin (BSA) as a standard. Mitochondria from MEF cells were isolated as described in ref. [56]. The activity of mitochondrial RC CIII was calculated from the slope of the increase in absorbance by cyt *c* (only the reduced form of cyt *c* absorbs at 550 nm). Data are expressed as the percentage of the control (i.e., the activity without the addition of PYO). To measure the activity of CIII and of PYO in vitro, isolated mitochondria (50 μg protein/ml) were resuspended in 50 mM potassium phosphate buffer, pH 7.5, supplemented with 10 μM alamethicin, 3 mg/ml BSA, 2.5 mM NaN₃, 2 μM rotenone, 2 μg/ml antimycin A, 0.025% Tween, and 75 μM oxidized cyt *c*. To start the reaction, reduced decylubiquinol (75 μM) was added followed by the addition of PYO at the indicated concentrations; absorbance changes were monitored at 550 nm and 37 °C.

**MTS assay.** To measure the viability of the cells, MEFs and human immortalized fibroblasts were seeded into 96-well plates at a density of $5 \times 10^3$ cells/well. After 24 h, the growth medium was replaced with Phenol Red- and FBS-free medium and treated with the drugs: four wells were used for each condition. After 8 h or 24 h of incubation, 10% CellTiter 96® AQUEOUS One solution (Promega, Italy) was

added to each well. Finally, absorbance at 490 nm was measured using an Infinite® 200 PRO 96-well plate reader.

**Determination of ATP concentration.** The amount of ATP was measured in MEFs and in human immortalized fibroblasts, using ATPlite™ Luminescence ATP Detection Assay System (PerkinElmer). Cells were seeded into white 96-well view plates at a density of $2 \times 10^4$ cells/well for MEF cells and $1.2 \times 10^4$ cells/well for human immortalized fibroblasts. To measure mitochondrial ATP production, 24 h after seeding glycolysis was inhibited pre-incubating the cells with 5.5 mM 2-DG for 1 h and then all treatments were performed in a medium containing 2-DG and no glucose. At 6 h after treatment, the assay was performed according to the manufacturer's instructions. To measure mitochondrial ATP production in *D. melanogaster*, flies were starved for 12 or 24 h prior to ATP content determination. Ten males flies per replicate and three biological replicates were used.

**Oxygen consumption assay.** Respiration was measured by using an XF24 Extracellular Flux Analyzer (Seahorse, Bioscience), which measures the oxygen consumption rate (OCR). Adherent MEFs or human fibroblasts were, respectively, seeded at $3 \times 10^4$ cells/well and $1.2 \times 10^4$ cells/well in 100 μl of DMEM and incubated for 24 h at 37 °C in a humidified atmosphere with 5% CO₂. The medium was then replaced with 670 μl/well of high-glucose DMEM without serum and sodium bicarbonate and supplemented with 1 mM sodium pyruvate and 4 mM L-glutamine. OCR was measured at preset time intervals upon the preprogrammed additions of the following compounds: oligomycin to 2 μg/ml, FCCP to 600 nM for MEFs and to 1.2 μM for human fibroblasts, and antimycin A to 1 μM final concentrations. All chemicals were added in 70 μl of DMEM. A massive loss of cells because of death and detachment was excluded by direct microscopic observation of the cells at the end of each experiment (not shown). Analysis of data was performed as described in ref. [57].

OCRs were analyzed also in *D. rerio* embryos, specifically in Ctrl MO; *TTC19* KD; and rescued *TTC19* KD. Respiratory competence of 72 hpf zebrafish embryos was assessed by Oxygraph+ (Hansatech instruments), O₂ concentration in the media, and O₂ consumption was measured using the O₂ view software. Fifteen randomly selected embryos were placed in the sample chamber (2 ml fish water, 28 °C, 50 r.p.m.), recording started after a stabilization period of 180 s, and acquisition was performed for 780 s. O₂ consumption/fish is shown in Fig. 7a, reporting 180 s of a mean plot obtained from six independent experiments, showing Ctrl MO; Ttc19 spMO; and 100 nM PYO pre-treated Ttc19 spMO.

Finally, mitochondrial respiratory function was measured in small samples of WT and *Ttc19*⁻/⁻ mice livers. Mouse livers were cut into samples of 20–30 mg and each sample was finely chopped using surgical scissors in ice-cold respiration medium MiR05, consisting of 3 mM MgCl₂, 0.5 mM EGTA, 60 mM lactobionic acid, 110 mM D-sucrose, 20 mM taurine, 10 mM KH₂PO₄, 20 mM HEPES, 1 g/L BSA (essentially fatty acid-free), and adjusted to pH 7.1 with KOH at 37 °C. Respiration was measured at 37 °C in 2-ml chambers of two-channel titration-injection respirometers (Oroboros Oxygraph, with DatLab software for data acquisition and analysis; Oroboros, Innsbruck, Austria). Instrumental background flux was assessed using MiR05 without biological samples.

**Mitochondrial membrane potential and ROS production.** MEF *Ttc19*⁺/⁻, MEF *Ttc19*⁻/⁻, and human fibroblasts, all at 80% of confluence, were collected from a 75 cm² flask and incubated in HBSS buffer (Thermo Fisher) for 30 min at 37 °C in the dark with 25 nM tetramethylrhodamine, methyl ester (TMRM) or 1 μM Mitosox®, respectively. Afterwards, the medium was diluted to maintain 5 nM TMRM as in Leanza et al. [55]. TMRM fluorescence was assessed at the indicated time points by fluorescence microscopy using a Leica SP5 fluorescence microscope equipped with the confocal setup (Leica Microsystem, Wetzlar, Germany). Antimycin A in combination with rotenone (both 2 μM) was used as a positive control for mitochondrial ROS production, while FCCP (2 μM) was used to completely dissipate mitochondrial membrane potential. Alternatively, after incubation with TMRM or Mitosox, cells were analyzed by a FACScanto II (BD Biosciences) and

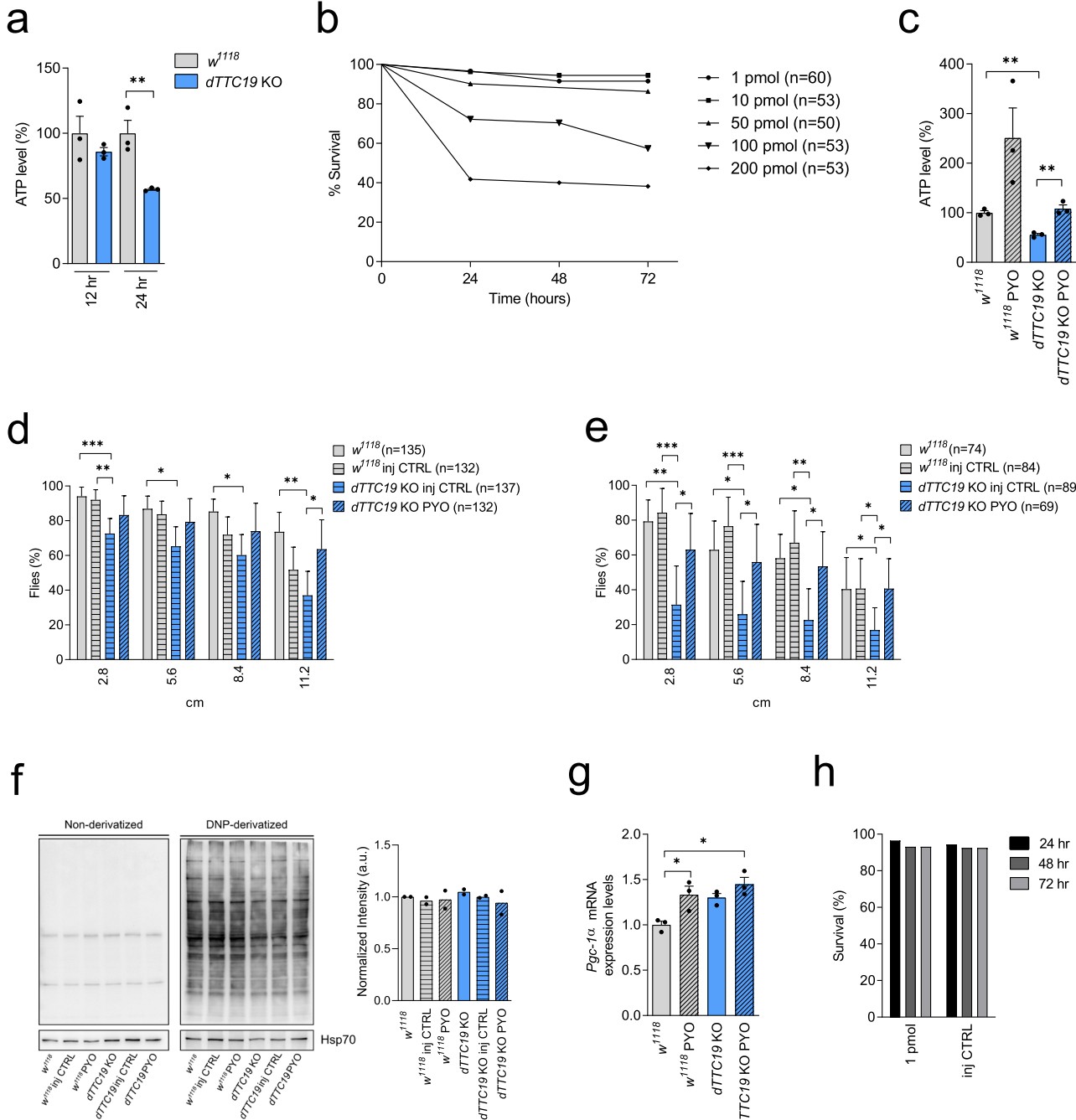

**Fig. 6 PYO-treated *dTTC19* KO flies. a** ATP in 5-day-old male flies (*dTTC19* KO) and controls (*w^1118^*), after 12–24 h fasting (*n* = 3 × 10 individuals). Significant difference between KO and controls was at 24 h. Means ± SEM (two-tailed Student's *t* test, **p < 0.01). **b** PYO toxicity assessed in WT flies (*w^1118^*) after 1.0–200 pmol PYO injection into the hemolymph of male flies. The % of surviving flies was recorded at 24, 48, and 72 h. Kaplan–Meier (*p* < 0.0001), with Bonferroni-corrected Mantel–Cox log-rank was: 1 pmol vs. 100–200 pmol: *p* < 0.001; 10 pmol vs. 100–200 pmol: *p* < 0.001; 50 pmol vs. 100–200 pmol: *p* < 0.01, *p* < 0.001 respectively. **c** ATP levels in 5-day-old control (*w^1118^* and *dTTC19* KO) and 1 pmol PYO (*w^1118^* PYO and *dTTC19* KO PYO) flies after 24 h fasting (*n* = 3 × 10 individuals). PYO increased ATP production in *dTTC19* KO flies to that of *w^1118^* control. Means ± SEM (two-tailed Student's *t* test, **p < 0.01). **d** PYO partially recovered bang sensitivity in 5-day-old flies (*dTTC19* KO PYO), vs. controls (*w^1118^*; *w^1118^* injection control flies, and *dTTC19* KO injection control flies). The % of flies reaching threshold distances (2.8, 5.6, 8.4, and 11.2 cm) is shown for the indicated genotypes, ±PYO injection. Means ± 95% CI (one-way ANOVA with post hoc on Tukey's test: *p < 0.05; **p < 0.01; ***p < 0.001). **e** As in (**d**). Bang test in 12-day-old KO flies after 1 pmol PYO (*dTTC19* KO PYO) vs. controls as above. The % of flies reaching the threshold distances are shown as above. Error bars mean 95% CI (one-way ANOVA with post hoc on Tukey's test: *p < 0.05; **p < 0.01; ***p < 0.001). **f** Representative oxyblot and quantification of protein oxidation with 1 pmol PYO (*w^1118^* PYO and *dTTC19* KO PYO) and relative controls as above (*n* = 2 independent experiments). **g** *Pgc-1α* expression assessed by qRT-PCR in *w^1118^* and *dTTC19* KO flies untreated or treated with 1 pmol PYO (*n* = 3 biological replicates). Means ± SEM (one-way ANOVA with post hoc on Tukey's test, *p < 0.05). **h** Toxicity associated with double injections (1 pmol PYO solution, *n* = 58 individuals vs. control solution, n = 53 individuals) assessed in WT male flies *w^1118^*. The % of surviving flies was assessed at 24, 48, and 72 h after the second injection.

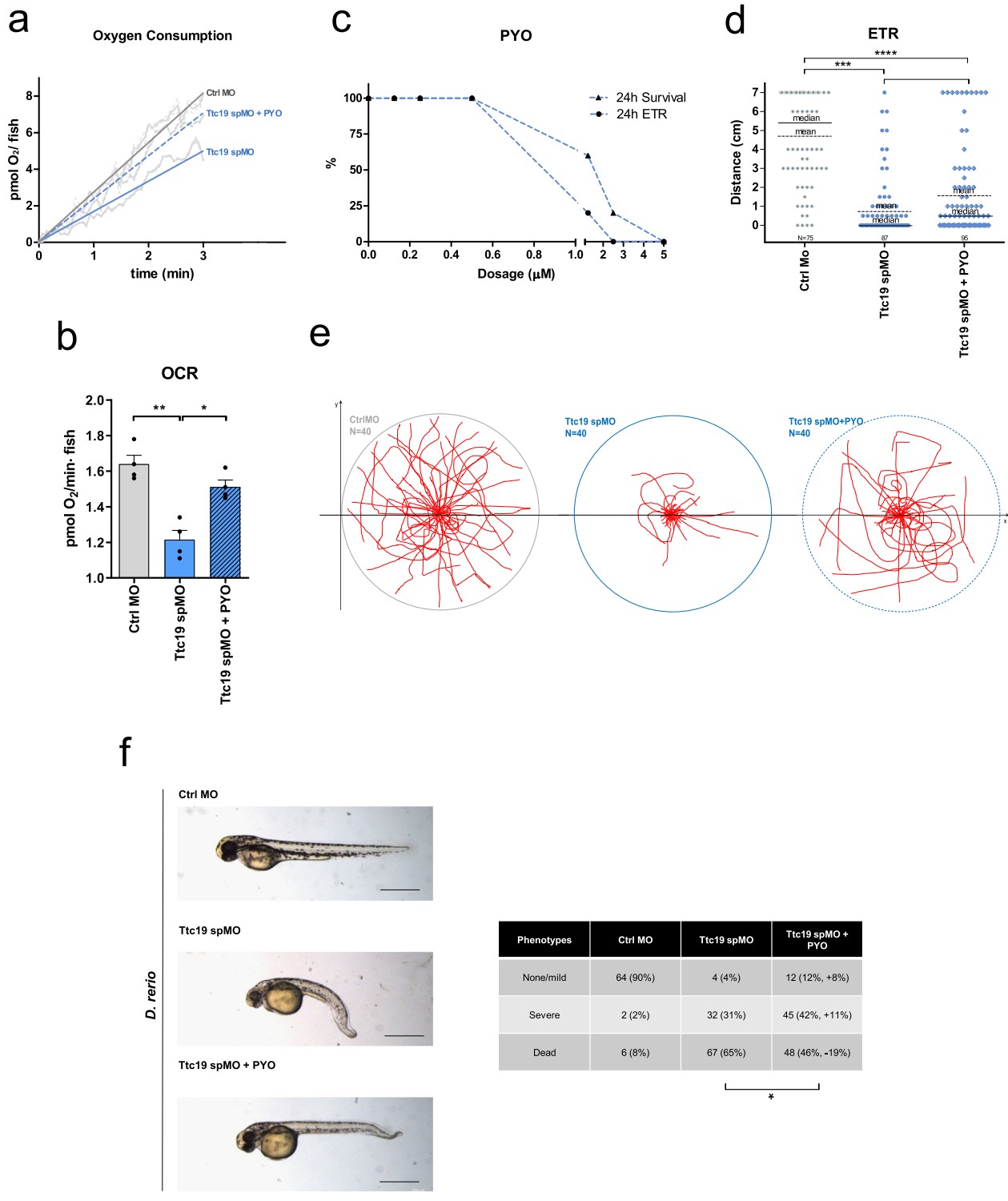

probes fluorescence was measured up to 60 min after PYO addition. A partial loss of TMRM fluorescence can also be observed in untreated samples. The data reported are the percentage of the median values of the fluorescence intensity distributions (10,000 cells were counted). Figures exemplifying the gating strategy are provided in the raw data section of the Supplementary material.

**Mitochondria morphology.** Transmission electron microscopy was performed as described in ref. [55]. Briefly, $1 \times 10^4$ MEFs and $5 \times 10^3$ fibroblasts were seeded in a 24-well plate. After 24 h, cells were treated for an additional 72 h, by changing the medium containing PYO or not every 24 h. After the treatment, samples

were fixed overnight, at 4 °C. Following washing, postfixation was performed and then sections were prepared for the analysis. Finally, the samples were observed with a Tecnai G2 Spirit transmission electron microscope (Fei electron microscopes) operating at 100 kV. Images were then analyzed using the ImageJ software, calculating the mean of all single mitochondrial areas of all the samples.

**Protein lysates.** MEF $Ttc19^{+/-}$ and MEF $Ttc19^{-/-}$, as well as human fibroblasts from healthy donor or patients, were seeded in standard 6-well plates at a density of $7 \times 10^4$ cells/well for MEFs or in 100 mm dishes at a density of $2.2 \times 10^5$ cells/

**Fig. 7 Characterization of *TTC19* KD zebrafish treated with PYO. a, b** OCR was analyzed in control morpholino (Ctrl MO), *TTC19* KD (Ttc19 spMO), and 100 nM PYO-rescued Ttc19 spMO zebrafish. Respiratory capability of zebrafish (72 h post fertilization (hpf)) was analyzed by oxygraph technology. $O_2$ consumption/fish is shown in graphs, including the best linear fit (quantified in **b**), reporting 180 s of a mean plot obtained from six independent experiments. Shown are data for Ctrl MO, Ttc19 spMO, and 100 nM PYO pre-treated Ttc19 spMO ($n = 6$ biological replicates with 15 randomly selected individuals for each replicate). SEM is reported as a dashed line for each group of fishes (**a**). $\Delta[pmolO_2]$/fish × min of treated fish are reported for Ctrl MO, *TTC19* KD, and rescue, respectively (**b**). Statistical analysis was performed by the two-tailed Student's *t* test. (\**p* < 0.05, \*\**p* < 0.01). Bars refer to SEM. **c** PYO dose–response survival assay was performed in embryos exposed to the indicated doses of the drug for 24 h. Vitality and positive reaction to evoked touch response (ETR) were assessed. **d** ETR assays were performed in 72 hpf zebrafish, to assess embryo movements by measuring the distance (cm) between the initial (center of the Petri dish) and the final position of the embryo measured as linear distance traveled starting from the center. Statistical analysis was performed by the two-tailed Student's *t* test. (\**p* < 0.05, \*\*\**p* < 0.001), means and medians are reported in the graph. The number of manipulated embryos is also reported. **e** Traces of movement of 40 individual fishes/Petri dish (circle) placed in the center of the dish and stimulated by touch. Fishes treated with 100 nM PYO (rightmost figure) were able to partially recover movement (see also Movie 1). **f** *TTC19* KD induces body shortening and tail malformation in comparison to sibling controls. One hundred nanomoles of PYO was administered in the fish water at 12 hpf, partially rescuing body malformation and in part death ($n = 6$ biological replicates, the precise number of analyzed zebrafish for each phenotype is reported in the figure). Scale bars: 0.5 mm. Statistical analysis was performed by the two-tailed Student's *t* test (\**p* < 0.05).

dish for fibroblasts, and allowed to attach overnight in complete DMEM medium. From the following day, cells were incubated with 1.5 or 0.8 μM of PYO (by changing the medium with freshly prepared one containing PYO), for MEFs and fibroblasts, respectively, or the corresponding amount of DMSO (vehicle). Incubation was carried out for 24, 48, or 72 h in standard DMEM with 2% of FBS. At the end of the treatment, the medium was removed and 150 μl of Lysis Buffer (25 mM Tris pH 7.8 + 2.5 mM EDTA + 10% glycerol + 1% NP40 + 2 mM dithiothreitol (DTT) + 1% phosphatase inhibitors cocktail 2 and 3 (Sigma-Aldrich) + 1% protease inhibitors cocktail (Sigma-Aldrich)) were added to each well. The plates were then frozen at −80 °C. Total extracts were collected from each well after scraping, vortexed for 10 s, and then centrifuged at $20,000 \times g$ for 10 min at 4 °C. To enhance protein separation, samples were solubilized for 30 min at room temperature (RT) in Sample Buffer (10% glycerol + 42 mM Tris/HCl pH 6.8 + 3% sodium dodecyl sulfate + 33 mM DTT + bromophenol blue).

**Western blotting.** Fifty micrograms of protein were loaded into each lane of 4–12% NuPage Bis-Tris precast gels (Thermo Fisher Scientific). The transfer was performed with NuPage Transfer Buffer. Membranes were blocked in 10% skim milk (Sigma-Aldrich) dissolved in Tris-buffered saline (TBS: 10 mM Tris, 150 mM NaCl, pH 7.4) for 1 h at RT. Membranes were incubated overnight at 4 °C with the following primary antibodies, all diluted in 5% skim milk prepared in TBS: Mfn2 (1:1000 in 5% skim milk—Abnova, H00009927-M03, clone4H8); ATP5A (1:1000 in 5% skim milk—Abcam AB14748 [15H4C4], lot: GR209582-7); β-actin (1:3000 in TTBS—EMD Millipore, MAB1501, clone C4, lot: 3166064); PGC-1α (1:1000 in TBS + 5% BSA—Abcam AB54481, lot: GR3315850-1); ND2 (1:500 in TTBS—Thermo Fisher Scientific, PA5-37185, lot: TC2524471C); SDHB (1:1000 in PBST + 3% BSA—Abcam, AB14714 [21A11AE7], lot: GR3272683-1); UQCRC2 (1:1000 in PBST + 3% BSA—Abcam, AB110252 [16D10AD9AH5], lot: GR3254174-5); MTCO1 (1:1000 in PBST + 3% BSA—Abcam, AB14705 [1D6E1A8]); SOD-1 (1:1000 in TTBS—Santa Cruz Biotechnology, sc11407 (FL-154)); CAT (1:1000 in TBS + 5% BSA—Abcam, AB1877); vinculin (1:2000 in TTBS—Millipore, AB6039, lot: 3215996); Na⁺K⁺ATPase (1:2000 in TTBS—Abcam, AB7671, clone 464.6); Hsp70 (1:5000 in PBST—Sigma-Aldrich, H5147, clone BRM-22). Anti-mouse (dilution 1:10,000 in TTBS; Bio-Rad, #1706516, lot: L005680 B) or anti-rabbit (dilution 1:10,000 in TTBS; Sera Care, 5220-0336, lot: 10283380) secondary antibodies conjugated to horseradish peroxidase were used for protein detection. Chemiluminescence was detected with a ChemiDoc Touch Imaging System (Bio-Rad) after incubation with Clarity Western ECL Substrate (Bio-Rad). Densitometric analyses were performed with the Image Lab software (Bio-Rad). β-Actin, vinculin, Na⁺K⁺ATPase, and Hsp70 were used as loading controls.

**Real-time PCR.** MEF $Ttc19^{+/-}$ and MEF $Ttc19^{-/-}$ were seeded in a T25 cm² flasks at a density of $1.8 \times 10^5$ cells/flask in a complete DMEM medium. From the following day, cells were incubated with 1.5 μM PYO or the corresponding amount of DMSO as control, for 48 or 72 h, in standard DMEM with 2% FBS. The medium was renewed every 24 h. At the end of the treatment, the medium was removed, and cells were washed once with PBS. Total RNA extraction was performed following the TRIzol (Invitrogen) extraction protocol, according to the manufacturer's instructions. Reverse transcription was performed using SuperScript II RT (Thermo Fisher Scientific), starting with 5 μg of RNA. Real-time quantitative polymerase chain reaction determinations/quantifications were then performed using Power SYBR® Green PCR Master Mix (Thermo Fisher Scientific) following the supplier's instructions. Reaction tubes were incubated first at 95 °C for 10 min. Forty cycles were carried out, with the following temperature protocol: 95 °C 15 s → 59 °C 10 s → 60 °C 50 s. iQiCycler (Bio-Rad) was used for the amplification reaction and Bio-Rad CFX Manager was used for plate reading and analysis. *Actin* was used as a housekeeping gene for normalization.

For flies, total RNA was extracted from ten adults using TRIzol reagent (Thermo Fisher Scientific). One microgram of total RNA was used for first-strand complementary DNA synthesis using SuperScript II (Life Technologies). Real-time quantitative PCRs were performed in triplicate using a Bio-Rad CFX 96 Touch System (Bio-Rad) using PowerUp SYBR Green chemistry (Thermo Fisher Scientific). The 2 −ΔΔCt (relative quantification) method was used to calculate the relative expression ratio[58]. Rp49 was used as an endogenous control. Three biological replicates, ten flies per replicate were used.

Sequences of all primers used are shown in Supplementary Table 1.

**Protein oxidation.** The protein oxidation level was measured using the OxyBlot™ Protein Oxidation Detection Kit (Millipore). MEF and human fibroblasts were seeded into 100 mm dishes at a density of $2.2 \times 10^5$ cells/dish or $2.4 \times 10^5$ cells/dish, respectively. From the following day, cells were treated with 1.5 or 0.8 μM PYO, respectively, for 3 days (renewing the medium every 24 h). After the treatment, the cells were collected, washed once with PBS, and the pellet was lysed in RIPA buffer (50 mM Tris-HCl pH 7.4, 150 mM NaCl, 0.25% deoxycholic acid, 1% NP40, 1 mM EDTA) supplemented with 50 mM DTT and 1% protease inhibitors cocktail (Sigma-Aldrich). The suspension was gently rocked for 15 min at 4 °C and then centrifuged at $14,000 \times g$ for 15 min at 4 °C.

To prepare protein samples from mice organs, samples were weighed and re-suspended in two volumes of RIPA buffer prepared as above. Liver, brain, and heart tissues were homogenized using TissueLyser II (Qiagen) for 3 min at 25 Hz. Samples were then centrifuged at $14,000 \times g$ for 10 min at 4 °C.

The protein content of the supernatant was determined by the Bradford protein assay. To prepare protein samples from flies, six individuals were homogenized in RIPA buffer prepared as above. Samples were then centrifuged at $14,000 \times g$ for 5 min at 4 °C in order to remove insoluble material. The protein content of the supernatant was determined by the Bradford protein assay. To derivatize the protein mixture, two aliquots of each sample were prepared with 20 μg of protein each. One aliquot was derivatized with 2, 4-dinitrophenyl hydrazine, and the other was treated with the derivatization-control solution, following the manufacturer's instructions. The derivatized protein samples were then separated by gel electrophoresis, followed by Western blotting. PVDF membranes were then saturated, incubated with primary and secondary antibodies provided by the Oxyblot™ Kit, and developed using chemiluminescent reagents as described in the manufacturer's protocol.

**Lipid peroxidation.** To measure lipid peroxidation in intact cells, MEFs and human fibroblasts were seeded into 96-well plates at a density of $1 \times 10^3$ cells/well or $0.5 \times 10^3$ cells/well, respectively. After 24 h, the cells were treated or not with PYO for 72 h as described above. Cells were also treated with 100 μM cumene hydroperoxide for 2 h, as a positive control. After treatment, the lipid peroxidation level was assessed using the Image-iT® Lipid Peroxidation Kit (Invitrogen). As recommended by the manufacturer, 10 μM Lipid Peroxidation Sensor was added into each well. After 30 min of incubation at 37 °C, cells were washed three times with PBS and then imaged with a Leica DMI4000 inverted microscope using a ×20 objective. The signal was then quantified using the Fiji software and the ratio between the red signal at 590 nm and the green one at 510 nm was used to quantify lipid peroxidation. In untreated negative control cells, most of the signal is in the red channel and the ratio is high. ROS-producing reagents lowered the ratios.

**Fly injection procedure and toxicity assessment.** Four-day-old male flies were anesthetized with $CO_2$ (for 5 min max), distributed on the pad, and injected using injection needles coupled to a cell injector (Narishige IM-300) and a micromanipulator (Leica)[59]. Unless otherwise specified, 1 pmol PYO diluted in Ringer's solution or Ringer's with Brilliant Blue FCF as tracking dye was injected

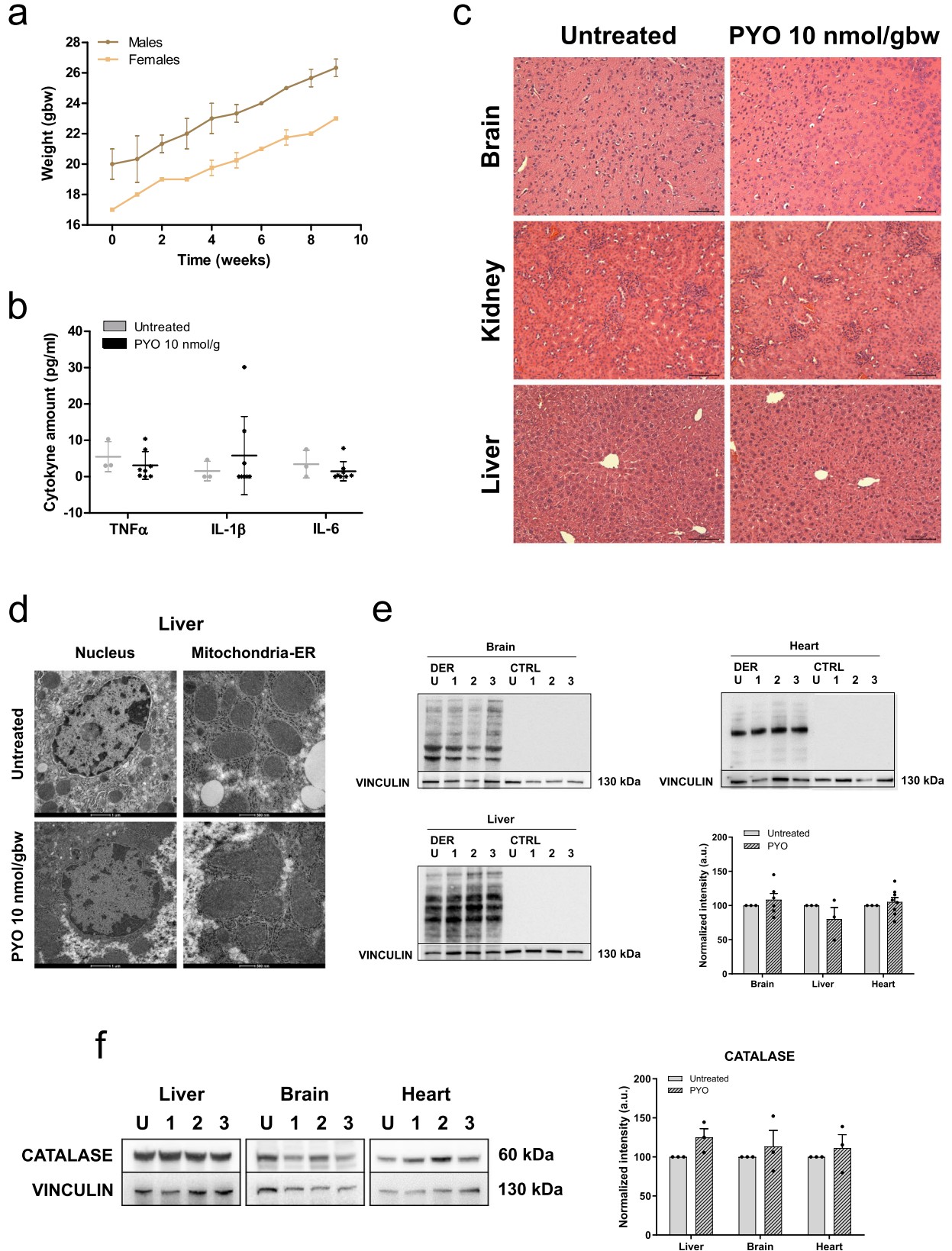

intrathoracically, at the slightly lighter-colored region between the mesopleura and pteropleura. As a control for injection (reported as "inj CTRL" in the text), Ringer's solution with Brilliant Blue FCF was used. PYO toxicity was previously assessed in WT flies ($w^{1118}$) after injection (from 1.0 to 200 pmol) in the hemolymph of male flies (50 individuals in groups of ten individuals). The percentage of surviving flies was evaluated at 24, 48, and 72 h after injection. During the evaluation of toxicity,

injection control flies were injected with the same volume of ethanol (solvent) used for PYO treatment, in order to exclude any effect of the solvent. To test toxicity associated with long-term treatment with low concentrations of PYO, flies were double injected. The second injection was performed 2 days after the first one. The percentage of surviving flies was evaluated at 24, 48, and 72 h after the second injection.

**Fig. 8 PYO showed no toxicity in vivo in wt mice. a** Effect of PYO (10 nmol/g) on mice body weight in 6-week-old WT C57Bl6/J males ($n = 4$) and C57Bl6/J females ($n = 4$). Weekly measurements of mice body weight (grams) are shown. Data are means ± SEM. **b** Plasma proinflammatory cytokine levels (TNF-α, IL-1β, IL-6) from mice treated with PYO 10 nmol/g ($n = 8$ independent samples) were not significantly altered compared to untreated controls ($n = 3$ independent samples). Bars indicate mean ± SEM with superimposed individual data points. **c** Representative light micrographs of the liver, kidney, and brain sections from mice untreated or treated with PYO (10 nmol/g) for 2 months ($n = 3$ biological replicates). All organs were free from gross pathological changes on hematoxylin and eosin (H&E) staining. Scale bar: 100 μm. **d** Representative TEM images of the liver sections from mice untreated or treated with 10 nmol/g PYO. Nucleus, mitochondria, ER, and the other detected organelles did not show any visible alterations ($n = 3$ biological replicates). Scale bars are reported in each picture. **e** Representative oxyblot of the liver, heart, and brain lysates from mice untreated (U) or treated (1, 2, and 3) with PYO (10 nmol/g) for 2 months. Samples were either derivatized by adding DNPH (2, 4-dinitrophenyl hydrazine) solution (DER) to visualize carbonyl groups introduced by oxidative modifications, or treated with derivatization-control solution (CTRL). Vinculin was used as a loading control (means ± SEM, $n = 3$ independent samples). **f** Representative Western Blot and the relative quantifications of catalase expression levels observed in lysates of liver, heart, and brain samples lysates from mice untreated (U) or treated (1, 2, and 3) with PYO (10 nmol/g) for 2 months. Vinculin was used as a loading control (means ± SEM, $n = 3$ independent samples). For all panels, statistical significance was calculated using two-tailed Student's $t$ test.

**Fly bang test**. After injection, flies were maintained on agar food for 24 h. On the following day, 1 h prior to the experiments, male flies were transferred without anesthesia to plastic vials (ten flies per vial), as described in ref. [11]. The test was performed 1 h after dark-light transition and a minimum number of 120 5-day-old individuals or 80 12-day-old flies were analyzed. Vials were mechanically stimulated by placement in a bench-top vortex for 10 s at the maximum setting. The vial containing the vortexed flies was then placed upright into a graduated cylinder in front of a digital webcam connected to a computer. The number of flies able to climb to predefined distances (2.8, 5.6. 8.4, and 11.2 cm) in 30 s was counted[60]. $w^{1118}$ and $dTTC19$ KO flies injected with Ringer's solution plus water were used as control. Non-injected $w^{1118}$ flies were used as a further control. Bang test sensitivity was assessed in KO flies after injection of 1 pmol of PYO ($dTTC19$ KO PYO) and compared to controls ($w^{1118}$; $w^{1118}$ injection control flies, and $dTTC19$ KO injection control flies) ($n \geq 120$ males per genotype).

**In vivo zebrafish embryos treatments**. Twenty WT zebrafish were arrayed in a 6-well plate and treated with either DMSO or PYO for 24 h, starting from 24 h.p.f. (endpoint: 48 hpf). Drugs were administered in 4 ml of fish water. Zebrafish embryos were analyzed using a Leica M165FC microscope. Survival was evaluated as the percentage of surviving fish. Three replicates were performed. For the methodologies of experiments on Wnt signalling see ref. [20].

**Touch-evoked escape response assays in zebrafish**. Touch-evoked escape response assays can be used to assess muscle performance and stimulus perception[61]. Zebrafish at 72 h.p.f. typically escapes after a tactile stimulation following in most cases a linear centrifugal path, reaching the border of 15 cm Petri disk, reflecting a defensive strategy against the stimulation source. Our method quantifies muscle performance by measuring the distance (cm) between the initial and the final position of the embryo. ETR was induced by a single and gentle stimulation at the tail of the larvae with a 200 μl thin polypropylene pipette tip. During a single short burst, the embryo moves to the final position. Video recordings were obtained using a standard video camera. Motility after stimulation was evaluated as the percentage of fishes able to move. Three replicates were performed, each involving >20 embryos. Data were processed by the Davinci Resolve software.

**In vivo mice treatments**. Six-week-old WT C57BL6/J (males and females) mice were injected i.p. with PYO diluted into physiological saline to reach the final dose of 10 nmol/g body weight. Injections were performed five times a week for a period of 2 months. The mice were then sacrificed and organs were collected and immediately frozen in liquid nitrogen. Organ samples were then used to detect the effects of PYO in vivo.

**Histology**. Thin sections (5 μm) of paraffin-embedded organ pieces were stained with a Hematoxylin and Eosin Staining Kit (BioOptica 04-061010) according to the manufacturer's instructions. Briefly, paraffin was removed with xylene, the tissue was rehydrated, washed with deionized water, stained first with hematoxylin, then with eosin, and finally dehydrated through serial ethanol dilutions and xylene. Glass coverslips were mounted over the samples using Entellan (Merck 1.07961.0100). Images were acquired using a Leica microscope.

**Enzyme-linked immunosorbent assay (ELISA)**. Tumor necrosis factor-α (TNF-α), interleukin-1β (IL-1β), and IL-6 concentrations from plasma samples were assessed by the standard ELISA technique (DuoSet ELISA, R&D Systems) following the manufacturer's instructions. The concentration was determined based on absorbance values of the recombinant mouse standard. Mouse TNF-α DY410 DuoSet ELISA, R&D Systems; mouse IL-1β DY401 DuoSet ELISA, R&D Systems; mouse IL-6 DY406 DuoSet ELISA, R&D Systems.

**PYO extraction from the liver**. Tissue levels of PYO were assayed in C57BL/6J mice. PYO was injected i.p. into mice, diluting a 5 mg/ml (i.e., 23.81 mM) stock solution in DMSO into 100 μl of physiological saline to reach the final dose of 10 nmol/g body weight. Thirty minutes after the injection, the liver was collected and immediately frozen. One hundred milligrams of tissue was weighed, PBS (1 vol) was added, and the tissue was homogenized using scissors and an electric pestle. A measure of 5 vol of chloroform was then added to the sample; after shaking for 5 min, the organic layer was collected. The remaining aqueous phase was extracted again with 5 vol of chloroform, as described above. The two organic phases were pooled, and 100 μl of 0.1 M HCl was added. After shaking for 5 min, the aqueous phase was collected and analyzed by HPLC/ultraviolet (UV).

HPLC analysis was performed with a 1290 Infinity LC System (Agilent Technologies) equipped with a UV diode array detector (190–500 nm), using a reverse-phase column (Zorbax SB-Aq, 1.8 μm, $2.1 \times 50$ mm$^2$; Agilent Technologies) kept at 35 °C. Solvents A and B were water containing 0.1% formic acid and acetonitrile containing 0.1% formic acid, respectively. The gradient for solvent B was as follows: B 0% for 0.5 min, from 0 to 25% in 2 min, from 25 to 100% in 2 min, and 100% for 1.5 min. The flow rate was 0.8 ml/min, and the injection volume was 2 μl. The eluate was preferentially monitored at 280 and 388 nm (corresponding to absorbance maxima for PYO). PYO was quantified using a calibration curve correlating peak area with the concentration of the analyte ($y = 5.4x$; $y$: peak area at 280 nm; $x$: concentration in μM units), and taking into account the mean recovery yield of PYO ensured by the extraction protocol. To determine the recovery yield, the protocol described above was applied to livers from untreated mice. A known amount of PYO was spiked into the tissue homogenate at the beginning of the procedure, and the recovery yield of PYO was determined as the ratio between the amount of PYO quantified in the final extract and the amount of PYO in the spike. The protocol ensured a recovery yield of 59.3 ± 7.6% from the liver (mean ± s.d., $n = 23$).

**Quantification and statistical analysis**. Statistically significant difference was assessed by means of one-way analysis of variance (ANOVA) analysis with Dunnett's post test, or two-way ANOVA for multiple comparisons. Two-tailed unpaired $t$ test analysis was used to compare the means of two groups. A result with a $p$ value of <0.05 was considered statistically significant. All statistical analyses were performed using the GraphPad Prism 9 software.

**Reporting summary**. Further information on research design is available in the Nature Research Reporting Summary linked to this article.

## Data availability
The authors declare that all the relevant data supporting the findings of this study are available within this article, its Supplementary information files, or are available from the corresponding author. Source data are provided with this paper.

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

## Acknowledgements

The work was funded by Telethon (Telethon grants to I.S. (GGP19118) and to RodC.), the NRJ Prix (Institut de France) and ERC advanced grant 322424 to M. Ze. S.C. was

supported by a post-doctoral fellowship no. BIRD182052 from the University of Padova (Italy), C.D.P. was supported by the grant "PRAT 2014—University of Padova, no. CPDA142980". T.V. was supported by the STARS project of the University of Padova. We are grateful to L. Scorrano, F. Di Lisa, and E. Gulbins for useful discussions and to D. Bonesso for help with pharmacokinetic experiments. We also thank the biobank "Cell line and DNA Bank of Genetic Movement Disorders and Mitochondrial Diseases" member of the Telethon Network of Genetic Biobanks (project no. GTB12001), funded by Telethon Italy, and of the EuroBioBank network, who provided us with specimens.

## Author contributions

Conceptualization: I.S., Rod.C., C.D.P.; methodology: R.P., S.C., Rob.C., M.B., M.Zo., C.V., T.V.; investigation: R.P., S.C., Rob.C., T.V., M.B., L.L., C.R., L.B., C.V.; resources: M.Zo., M.Ze., C.D.P., D.G.; writing—original draft: I.S., S.C.; editing: M. Ze., M. Zo., Rod. C., R.P., Rob.C., M.B., C.V.; funding acquisition: I.S., Rod.C., C.D.P., M. Ze.; supervision: I.S., Rod.C., C.D.P., C.V.

## Competing interests

I.S., R.P., R.C., M. Ze and M. Zo are inventors of a patent application to use pyocyanin and related compounds to treat mitochondrial diseases filed by the University of Padova (No. 102021000006065). All other authors declare no competing interests.
