## [Peer Review File · Nature Communications]

Reviewers' Comments:

Reviewer #1:

Remarks to the Author:

In the manuscript entitled "Exploiting pyocyanin to treat mitochondrial disease due to respiratory complex III dysfunction", the authors show that CIII function in electron transport can be bypassed by pyocyanin supplementation in an array of CIII deficiency models. Convincingly, PYO increases respiration, mitochondrial membrane potential and ATP production in MEF cells, when CIII activity is compromised. However, in TTC19 deficient human fibroblasts, the increase in respiration associated with PYO treatment does not lead to an increase in mitochondrial membrane potential (Fig 3H, right panel). The question of whether, in patient fibroblasts, PYO treatment enhances energy output (e.g. ATP production), remains open. Moreover, the authors report an increase in mitochondrial biogenesis upon PYO supplementation in MEF and human fibroblasts. I believe that it would be important to determine CIII steady-state levels in these cells. An induction of mitochondrial biogenesis could lead to an increase in the accumulation of the residual CIII, which, even if only partially functional, could contribute to the observed improvements in mitochondrial respiration. Lastly, in my opinion, the authors dismiss too quickly, as non-toxic, the increase in ROS production measured after 24 hours of PYO supplementation. At least, they should directly measure some markers of oxidative stress as protein or lipid oxidation or induction of antioxidant defenses. Moreover, at this point, we cannot exclude that a constant long-term increase in ROS levels of ~20% could have negative effects and caution should be exercised when proposing PYO as potential therapeutic treatment.

Minor points:

- In the way in which the data are presented, it is not possible to understand whether CIII deficient cells present an increase in ROS basal level (untreated) compared to control.
- In page 3, the authors state that "3 uM or lower PYO concentrations did not compromise cell survival and metabolism (Fig 1A)". Figure 1A reports exclusively data regarding cell viability. To which data are the authors referring to regarding metabolism?

Reviewer #2:

Remarks to the Author:

The paper by Peruzzo et al. reports on investigation of pyocyanin as a potential treatment for mitochondrial disease due to respiratory complex III dysfunction. The authors show that pyocyanin, a bacterial redox cyler, is able to replace complex III function by providing an electron shunt pathway in very low non-toxic concentrations. Administration of this drug restored respiration and increased ATP production in fibroblasts from patients with mutations in TTC19, an assembly factor of complex III. These findings were supported by studies in *Drosophila* and zebrafish models of TTC19 deficiency.

I find the paper interesting however there are some issues which need clarification, as the supporting evidence is not satisfactory. My worry is that this compound would not be easily administered in humans. Treating fish/cells/flyes for a short period of time does not constitute a problem however even low amounts of ROS species for sustained periods of time can have a more severe effect.

1. I think the choice of cell models is not optimal as we know that fibroblasts often do not show defects of OXPHOS components. It is not easy to understand whether these MEF cells actually express the complex III defect in the paper. Since the authors have the mouse model they could have tried to treat cells derived from affected organs, which more clearly show the complex III defect.
2. It would be also nice to see some WB of the complexes in treated cells in order to establish the phenotype of these cells.
3. I find it a bit confusing that the control and affected cells are in different graphs in more experiments in Figure 1. It is hard to assess the changes between their control and disease model.
4. How did the authors quantify fragmentation?

5. In order to support that they see an increase in mitochondrial biogenesis it would be nice to see WB of PGC1alpha or determination of mtDNA copy numbers.
6. Figure 5 (zebrafish) Is the compound added immediately after injection? How many days post fertilization are the fish kept in this compound?
7. The fish seem to have a muscle/NMJ or PNS phenotype. It would be nice to show it (eg fast/slow muscle fibre, NMJ:SV2 and bungarotoxin staining) if possible, however it is not essential.
8. The authors do not provide evidence regarding the beneficial effects of the ROS generated due to this compound. It may be possible to investigate if the amount of ROS generated could lead to a short-term mitochondrial stress response, which in turn would be beneficial.

Reviewer #3:

Remarks to the Author:

The manuscript by Peruzzo et al. described the use of a small molecule (Pyocyanin), a bacterial redox cyler, to compensate for a complex III defect. Pyocyanin has been studied as a potent toxin, a virulent factor from *Pseudomonas aeruginosa*. However, the authors found that when using sub-toxic concentrations, Pyocyanin acted as a mild ROS producer and induced an increase in ATP levels, not only in complex III defective, but also in control cells. The authors also found that Pyocyanin stabilized mitochondrial membrane potential and enhanced organelle biogenesis. Besides MEF and human fibroblasts, they also found these effects in complex III deficient *Drosophila* (TTC19KO) and in zebrafish (TTC19KD). Low concentrations of pyocyanin improved movement phenotypes.

1- I found the study very interesting and of high translational potential. However, a lot more needs to be done to assure the absence of toxicity. On this line, I was surprised that Pyocyanin was not tried in the TTC19 mouse model. The authors used MEF cells from the model, but not the mice, preferring to use zebrafish and *Drosophila*.

Other concerns:

2- The authors state that "indicate that PYO accepts electrons from decylubiquinol and reduces cytochrome c, ". It could reduce cyt c and not necessarily accept electrons from decylubiquinol.

3- The conclusion that "incubation of the cells with 0,8 or 1.5 μ M PYO increased ATP production by approximately two-fold." Is difficult to understand. ATP production is a consequence of proton pumping by complexes I, III and IV. How is PYO increasing the electrochemical gradient in WT cells?

4- PGC-1a could be studied by western blot

5- I do not see the concentrations, experiment of rescuing Wnt signaling reduction in CIII-deficient MEFs in Figure 2F.

6- The differences in GFP look extremely small in Figure 2G (even if significant). Is that meaningful?

7- I do not see error bars or significance stars of figure 4 D and 4E

8- It would have been interesting looking at a different complex III defect model (e.g. *Bcsl1L*) to test the generality of the approach.

Reviewer #1 (Remarks to the Author):

In the manuscript entitled “Exploiting pyocyanin to treat mitochondrial disease due to respiratory complex III dysfunction”, the authors show that CIII function in electron transport can be bypassed by pyocyanin supplementation in an array of CIII deficiency models. Convincingly, PYO increases respiration, mitochondrial membrane potential and ATP production in MEF cells, when CIII activity is compromised.

We thank the Reviewer for appreciating our work and useful suggestions. We took into account all indications and performed a series of new experiments to answer the questions.

However, in TTC19 deficient human fibroblasts, the increase in respiration associated with PYO treatment does not lead to an increase in mitochondrial membrane potential (Fig 3H, right panel).

We thank the Reviewer for calling our attention to this point. We repeated this experiment as well as those regarding ATP production in freshly thawed fibroblasts from the same patient (shown in Fig. 4G of the current version) and concluded that most probably the cells were senescent when used for the experiment of former Figure 3H, we apologize for this mistake. To confirm that the effect we observe with the freshly thawed cells is indeed a reliable and general observation, we performed the measurements on fibroblasts from two other TTC19 mutant patients. In all cases, PYO was not only able to stabilize the membrane potential (Figures 4G and S3C), but also increased respiration (Figure 4D,E and S3B), ATP production (Figure 4C) and ROS production (Figures 5B and S3E). Thus, all bioenergetic effects we observed with sub-lethal doses of PYO were confirmed in cells of three independent patients.

In addition, following the suggestion of Reviewer 3, we extended our observations to cells derived from patients harbouring mutations in BSC1L or LYRM7, two factors required for correct assembly/function of CIII. As shown in Figure S4B, the mitochondrial ATP level in LYRM7-mutated fibroblasts is lower than that measured in healthy fibroblasts and upon addition of sub-lethal dose of PYO, ATP production is completely recovered in the mutant fibroblasts. Again, stimulated respiration increased (Figure S4C), the membrane potential was stabilized (Figure S4D) and a mild ROS production was triggered (Figure S4D). The same set of experiment leading to comparable results were performed with fibroblasts of a patient harbouring biallelic BSC1L mutation (Figure S5 A-E).

Altogether, these results, showing that PYO exerts the same effect in fibroblasts of five different CIII-defective patients due to three different gene defects, points to high reproducibility of its mode of action, independently of the nature of the mutation leading to impaired complex III function that decreased mitochondrial ATP level in all examined cases.

The question of whether, in patient fibroblasts, PYO treatment enhances energy output (e.g. ATP production), remains open.

As mentioned above, now we provide evidence that PYO, at sub- μ M concentration is able to enhance ATP production by 50-100% in fibroblasts from 5 different CIII-defective patients harbouring mutations in different genes. In particular, in cases where the mitochondrial ATP level is decreased by approximately 30%, PYO can completely restore the ATP level. However, also in the case of the BSC1L where mitochondrial ATP level is drastically reduced by more than 60% (Figure S5B), PYO is able to exert a positive effect, doubling mitochondrial ATP content.

Moreover, the authors report an increase in mitochondrial biogenesis upon PYO supplementation in MEF and human fibroblasts. I believe that it would be important to determine CIII steady-state levels in these cells. An induction of mitochondrial biogenesis could lead to an increase in the

accumulation of the residual CIII, which, even if only partially functional, could contribute to the observed improvements in mitochondrial respiration.

The significantly lower steady state level of CIII has been previously determined in various human fibroblasts bearing CIII-linked mutations in comparison to the healthy ones (Ghezzi et al, Nat. Gen. 2011 for TTC19; Dallabona et al, Brain, 2016 for LYRM7; Fernandez-Vizzarra, Human Mol Gen. 2007 for BCS1L), and are confirmed here for the cell line mostly used in our study, MEF TTC19 (Figure 1E). Decreased complex III activity was also confirmed (Figure 1F). While we agree with the Reviewer that mitochondrial biogenesis could enhance the accumulation of partially functional CIII and indeed CIII protein level was found to be slightly increased after 72 hours of incubation (Figure S2F), we underscore that increase in respiration is instantaneous (Fig. 1C). This rules out that increased mitochondrial biogenesis may uniquely account for the observed improvement in mitochondrial respiration.

Lastly, in my opinion, the authors dismiss too quickly, as non-toxic, the increase in ROS production measured after 24 hours of PYO supplementation. At least, they should directly measure some markers of oxidative stress as protein or lipid oxidation or induction of antioxidant defenses. Moreover, at this point, we cannot exclude that a constant long-term increase in ROS levels of ~20% could have negative effects and caution should be exercised when proposing PYO as potential therapeutic treatment.

We are grateful to the Reviewer for pointing out this important issue. We performed the requested experiments, both *in vitro* and *in vivo*. In particular, ROS level failed to induce lipid peroxidation, either after 72 hours of incubation (Figure 2B and Figure S2A) or after culturing TTC19^{-/-} MEFs for 2 months in the presence of PYO (Figure 2C and Figure S2B). Likewise, protein oxidation was not increased after incubation with PYO (Figure 2D and Figure S2C for 72 h, and Figure 2E for 2-month treatment). Accordingly, no significant up-regulation of the anti-oxidant enzyme catalase occurred even after long-term incubation with PYO (Figure 2F). Similar results, including lack of protein oxidation, were confirmed also in the three patient-derived TTC19 mutant fibroblasts (Figure 5C,D,E) as well as in those from the BCS1L mutant individual (Figure S5F-G). Interestingly, catalase activity was somewhat enhanced in the BCS1L case after 72 hours of incubation with PYO, but SOD levels remained the same. For the TTC19 patients, lack of lipid peroxidation was also determined (Figure S3F).

For the *in vivo* part, we saw no increase in protein oxidation in *Drosophila* following PYO administration (Figure 6F). Most importantly, we treated adult mice with PYO (10 nmol/gbw, i.p.) once a day, 5 days a week, for two months (n=8). No arrest in growth (Figure 8A) or inflammatory phenotype (Figure 8B) occurred in these animals. Different tissues were examined by histology showing no gross alterations (Figure 8C). Furthermore, ultrastructure of nucleus and mitochondria in liver (Figure 8D) as well as in brain and muscle (Fig S6B) was not affected in these wild-type mice. Protein oxidation level (Figure 8E) as well as CAT expression levels (Figure 8F) were not significantly altered in liver, brain and heart of animals treated with PYO. To verify that PYO is able to reach target organs after i.p. administration, we measured its levels in the liver over a 4-hour period after a single injection. PYO was present, at concentrations < 1 nanomole/gbw (n=3)). Altogether, these new data indicate that low-dose PYO administration is not toxic, even at a long time-scale (2 months).

Minor points:

- In the way in which the data are presented, it is not possible to understand whether CIII deficient cells present an increase in ROS basal level (untreated) compared to control.

Unfortunately the Mitosox probe does not allow to reliably compare the basal ROS level among different cells, but please note that the basal lipid or protein peroxidation profile is comparable in MEF TTC19^{+/-} cells versus the MEF TTC19^{-/-} cells (Figures 2B-E) and the antioxidant enzyme catalase also displays comparable expression levels (Figure 2F). The same considerations stand for human fibroblasts from patients with TTC19 (Figures 5 C-E) or BSC1L (Figure S5F) mutations.

- In page 3, the authors state that “3 uM or lower PYO concentrations did not compromise cell survival and metabolism (Fig 1A)”. Figure 1A reports exclusively data regarding cell viability. To which data are the authors referring to regarding metabolism?

We referred to the fact that respiration was not compromised. Now we deleted the word metabolism from the sentence.

Reviewer #2 (Remarks to the Author):

The paper by Peruzzo et al. reports on investigation of pyocyanin as a potential treatment for mitochondrial disease due to respiratory complex III dysfunction. The authors show that pyocyanin, a bacterial redox cyler, is able to replace complex III function by providing an electron shunt pathway in very low non-toxic concentrations. Administration of this drug restored respiration and increased ATP production in fibroblasts from patients with mutations in TTC19, an assembly factor of complex III. These findings were supported by studies in Drosophila and zebrafish models of TTC19 deficiency.

I find the paper interesting however there are some issues which need clarification, as the supporting evidence is not satisfactory. My worry is that this compound would not be easily administered in humans. Treating fish/cells/flies for a short period of time does not constitute a problem however even low amounts of ROS species for sustained periods of time can have a more severe effect.

We thank the Reviewer for the suggestions that allowed us to improve our work, also by adding many new experiments, as indicated below. Regarding the administration in humans, we hope that one day, hopefully soon, it will be possible, but for the moment, our study is a proof-of-principle study; of course we cannot exclude that other redox cyclers may prove to be more suitable for therapy. In order to test long-term toxicity, we treated adult mice with PYO (10 nmol/gbw, i.p.) once a day, 5 days a week, for two months (n=8). No arrest in growth (Figure 8A) or inflammatory phenotype (Figure 8B) occurred in these animals. Different tissues were examined by histology showing no gross alterations (Figure 8C). Furthermore, ultrastructure of nucleus and mitochondria in liver (Figure 8D) as well as in brain and muscle (Fig S6B) was not affected in these wild-type mice. Protein oxidation level (Figure 8E) as well as CAT expression levels (Figure 8F) were not significantly altered in liver, brain and heart of animals treated with PYO. To verify that PYO is able to reach target organs after i.p. administration, we measured its levels in the liver over a 4-hour period after a single injection. PYO was present, at concentrations < 1 nanomole/gbw (n=3)). Altogether, these new data indicate that PYO administration is not toxic, even at a long time-scale (2 months).

I think the choice of cell models is not optimal as we know that fibroblasts often do not show defects of OXPHOS components. It is not easy to understand whether these MEF cells actually express the complex III defect in the paper. Since the authors have the mouse model they could have tried to treat cells derived from affected organs, which more clearly show the complex III defect.

The MEFs used in this study have previously been characterized (Ghezzi et al, 2011; Bottani et al 2017) and were kindly gifted by the authors of these papers. Indeed, as expected, Figure 1E of the

original ms. showed decreased respiration in TTC19^{-/-} MEFs with respect to control fibroblasts. In the current version of the manuscript, we confirmed the reduced quantity and activity of CIII (Figures 1E and F of the revised version) in TTC19 mutant MEFs. The Reviewer is correct that Prof. Zeviani, who is among the authors has the TTC19 mice model. Unfortunately, since he moved to Padova from Cambridge only recently, he could transfer only a few TTC19 mice here for the moment, and the block of colony expansion during the COVID lock-down impeded us to establish yet a mouse colony suitable to carry out statistically significant experiments for complex behavioural studies. Nevertheless, we took advantage of the few mice we had and assessed whether PYO may exert a positive effect also in a tissue relevant for the disease, as kindly suggested by the Reviewer. To this end, we isolated liver from WT and TTC19^{-/-} mice (n=3 for each group) and assessed respiration by high resolution oxygraphy (Oroboros Oxygraph). As expected, oxygen consumption, which was reduced in TTC19^{-/-} versus TTC19^{+/-} liver (Figure 1I), was increased by PYO, in both basal conditions (Figure 1J) and in the presence of the CIII inhibitor Antimycin A (Figure 1K). However, the increased oxygen consumption was sensitive to complex IV blockers (Figure 1J,K), confirming that PYO shuttled electrons predominantly to cytochrome c rather than directly to oxygen. Thus, the benefit of PYO administration on respiration is confirmed also in a tissue derived from an affected organ (Bottani et al, 2017, Molecular Cell).

2. It would be also nice to see some WB of the complexes in treated cells in order to establish the phenotype of these cells.

We included these new data into the revised manuscript, revealing a tendency to enhance the level of proteins belonging to respiratory chain complexes I-V (Fig 3C, Fig S2F), and further suggesting enhanced mitochondrial biogenesis.

3. I find it a bit confusing that the control and affected cells are in different graphs in more experiments in Figure 1. It is hard to assess the changes between their control and disease model.

Many thanks for this useful comment, in the revised version we merged the graphs for the control and diseased cells in Figures 1A, 4A, 4B, and in all new figures.

4. How did the authors quantify fragmentation?

TEM images were analyzed using ImageJ software, calculating the mean of the mitochondria areas of all the samples (Figures 3A for TTC19 MEFs and 5H for BSC1L human fibroblast).

5. In order to support that they see an increase in mitochondrial biogenesis it would be nice to see WB of PGC1alpha or determination of mtDNA copy numbers.

We now confirmed also in Western Blots that PYO triggers an increase in PGC1 α level (Figures 3F and S5G).

6. Figure 5 (zebrafish) Is the compound added immediately after injection? How many days post fertilization are the fish kept in this compound?

We apologize for the inaccuracy of the description of the methods. We now corrected the sentence, adding in the methods section "The compound was added between 8 hpf and 8.5 hpf, immediately after chorion punching with a thin needle".

7. The fish seem to have a muscle/NMJ or PNS phenotype. It would be nice to show it (eg fast/slow muscle fibre, NMJ:SV2 and bungarotoxin staining) if possible, however it is not essential.

We thank the reviewer for the suggestion. Since we have a zebrafish reporter line, which highlights motoneuron development and elongation *in vivo* without the need of chemical staining with Bungarotoxin, we used this reporter line (mnx1) to check Peripheral Nervous System. We observed impaired development in Ttc19spMO at 36 hpf (Figure S6A). This phenotype is compatible with the role of Wnt signalling in neuromuscular junction proposed by Strochlic et al., (Plos One, 2012). In our experiments Pyocyanin did not improve mnx1 expression or enhance the ratio motoneurons/area, but favoured motoneuron axon elongation in PYO treated animals versus DMSO treated Ttc19spMO. Wnt signalling recovery (current Figure 3H) may partially explain axon elongation, but we cannot state if the effect is fully cell autonomous or is due to cell-cell communication (e.g. axon guidance).

8. The authors do not provide evidence regarding the beneficial effects of the ROS generated due to this compound. It may be possible to investigate if the amount of ROS generated could lead to a short-term mitochondrial stress response, which in turn would be beneficial.

We believe that the beneficial effects of PYO-triggered mild ROS release are appreciable by the increased level of PGC-1 α and enhanced mitochondrial biogenesis observed in various systems (cells, Drosophila). On the other hand, PYO-induced increase in respiration and mitochondrial ATP level is also beneficial.

Reviewer #3 (Remarks to the Author):

The manuscript by Peruzzo et al. described the use of a small molecule (Pyocyanin), a bacterial redox cyler, to compensate for a complex III defect. Pyocyanin has been studied as a potent toxin, a virulent factor from *Pseudomonas aeruginosa*. However, the authors found that when using sub-toxic concentrations, Pyocyanin acted as a mild ROS producer and induced an increase in ATP levels, not only in complex III defective, but also in control cells. The authors also found that Pyocyanin stabilized mitochondrial membrane potential and enhanced organelle biogenesis. Besides MEF and human fibroblasts, they also found these effects in complex III deficient *Drosophila* (TTC19KO) and in zebrafish (TTC19KD). Low concentrations of pyocyanin improved movement phenotypes.

1) I found the study very interesting and of high translational potential. However, a lot more needs to be done to assure the absence of toxicity. On this line, I was surprised that Pyocyanin was not tried in the TTC19 mouse model. The authors used MEF cells from the model, but not the mice, preferring to use zebrafish and *Drosophila*.

We thank the Reviewer for the positive evaluation of our work and for the useful suggestions. We took into account all indications and performed a series of new experiments to answer the questions. In order to test long-term toxicity, we treated adult mice with PYO (10 nmol/gbw, i.p.) once a day, 5 days a week, for two months (n=8). No arrest in growth (Figure 8A) or inflammatory phenotype (Figure 8B) occurred in these animals. Different tissues were examined by histology showing no gross alterations (Figure 8C). Furthermore, ultrastructure of nucleus and mitochondria in liver (Figure 8D) as well as in brain and muscle (Fig S6B) was not affected in these wild-type mice. Protein oxidation level (Figure 8E) as well as CAT expression levels (Figure 8F) were not significantly altered in liver, brain and heart of animals treated with PYO. To verify that PYO is able to reach target organs after i.p. administration, we measured its levels in the liver over a 4-hour period after a single injection. PYO was present, at concentrations < 1 nanomole/gbw (n=3)).

Altogether, these new data indicate that PYO administration is not toxic, even at a long time-scale (2 months).

Other concerns:

2) The authors state that “indicate that PYO accepts electrons from decylubiquinol and reduces cytochrome c, “. It could reduce cyt c and not necessarily accept electrons from decylubiquinol.

The reviewer is correct, PYO could accept electrons from NADH. We indeed checked that *in vitro* PYO was able to accept electrons: Figure S1B shows that PYO can accept electrons, at least *in vitro*, also from NADH, as expected. However, in intact cells NADH depletion does not seem to occur, since the extracellular acidification rate reflecting glycolysis does not decrease (Figure S1C). Independently of the source of electrons for PYO, the important aspect is that it is able to reduce cytochrome c.

3- The conclusion that “incubation of the cells with 0,8 or 1.5 μ M PYO increased ATP production by approximately two-fold.” Is difficult to understand. ATP production is a consequence of proton pumping by complexes I, III and IV. How is PYO increasing the electrochemical gradient in WT cells?

We thank the Reviewer for calling our attention to this point: we have now changed the text and replaced “ATP production” with “mitochondrial ATP level”, since indeed this is the parameter that we measured. Mitochondrial ATP level depends on the ratio between the rates of ATP production and of ATP consuming reactions. When ATP production increases due to enhanced respiration without a concomitant increase in ATP consumption, this might elevate the overall ATP level. Regarding the possible mechanism by which PYO enhances ATP levels in mitochondria of WT cells, we tried to understand whether the action of PYO may be similar to that of another redox cyler, Methylene Blue (MB). MB was shown to enhance ATP level in isolated WT mitochondria, presumably due to substrate level phosphorylation (Komlodi and Tretter 2017). However our results shown in Figure 2I are not fully compatible with such mechanism as they show that CIV is required for PYO to maintain membrane potential. Therefore, further work will be needed to understand if PYO may have some additional effect e.g. on ATP consumption.

4- PGC-1 α could be studied by western blot

We now confirmed also in Western Blots that PYO triggers an increase in PGC1 α level (Figures 3F and S5G).

5- I do not see the concentrations, experiment of rescuing Wnt signaling reduction in CIII-deficient MEFs in Figure 2F.

We apologize, this has been corrected in the revised figure legend.

6- The differences in GFP look extremely small in Figure 2G (even if significant). Is that meaningful?

When working with whole zebrafish, such a difference is generally considered significant and meaningful. In any case, now we included a further experiment that was suggested by Reviewer 2 (Figure S6A) whose result is compatible with enhanced Wnt signalling. We exploited a zebrafish reporter line, which highlights motoneuron development and elongation *in vivo* (mnx1). We used this reporter line to check Peripheral Nervous System. We observed a development impairment in Ttc19spMO at 36 hpf (Figure S6A). This phenotype is compatible with the role of Wnt signalling in

neuromuscular junction proposed by Strohlic et al., (Plos One, 2012). In our experiments PYO favoured motoneuron axon elongation. Wnt signalling recovery (current Figure 3H) may partially explain axon elongation.

7- I do not see error bars or significance stars of figure 4 D and 4E

We apologize for this, the experiment shown derived from the indicated numbers of flies that were treated and investigated in groups of 10 individuals. Statistical analysis was performed and is included in the text and legend.

8- It would have been interesting looking at a different complex III defect model (e.g. Bcs1L) to test the generality of the approach.

We have complied and extended our observations to cells derived from patients harbouring mutations in BSC1L or LYRM7, two factors required for correct assembly/function of CIII. As shown in Figure S4B, the ATP level in LYRM7-mutated fibroblasts is lower than that measured in healthy fibroblasts and upon addition of sub-lethal dose of PYO, ATP production is completely recovered in the pathologic fibroblasts. In addition, also in this case stimulated respiration increased (Figure S4C), the membrane potential was stabilized (Figure S4D) and a mild ROS production was triggered (Figure S4D). The same set of experiment leading to comparable results were performed with fibroblasts of a patient harbouring BSC1L mutation (Figure S5 A-E).

Reviewers' Comments:

Reviewer #1:

Remarks to the Author:

In this revised version of the manuscript by Peruzzo et al, the authors have included a remarkable amount of additional experimental work that strengthens the conclusion of the study and addresses the previous concerns raised by the reviewers.

Reviewer #2:

Remarks to the Author:

I have no more comments. I think the manuscript has improved.

Reviewer #3:

Remarks to the Author:

The authors improved the manuscript. Legends to supplementary figures need careful review. For example, Fig. S2A-B is missing ".....of lipid peroxidation detection in..."